# SIGNATURES MEET DYNAMIC PROGRAMMING: GENERALIZING BELLMAN EQUATIONS FOR TRAJECTORY FOLLOWING

## ABSTRACT

Path signatures have been proposed as a powerful representation of paths that efficiently captures the path's analytic and geometric characteristics, having useful algebraic properties including fast concatenation of paths through tensor products. Signatures have recently been widely adopted in machine learning problems for time series analysis. In this work we establish connections between value functions typically used in optimal control and intriguing properties of path signatures. These connections motivate our novel control framework with signature transforms that efficiently generalizes the Bellman equation to the space of trajectories. We analyze the properties and advantages of the framework, termed signature control. In particular, we demonstrate that (i) it can naturally deal with varying/adaptive time steps; (ii) it propagates higher-level information more efficiently than value function updates; (iii) it is robust to dynamical system misspecification over long rollouts. As a specific case of our framework, we devise a model predictive control method for path tracking. This method generalizes integral control, being suitable for problems with unknown disturbances. The proposed algorithms are tested in simulation, with differentiable physics models including typical control and robotics tasks such as point-mass, curve following for an ant model, and a robotic manipulator.

## 1 INTRODUCTION

Dynamic programming (DP) is a foundation of many of the modern decision making algorithms such as optimal control (Liberzon, 2011) and reinforcement learning (RL) (Sutton & Barto, 2018). Typically, dynamic programming (Bellman, 1953) over the scalar or *value* of a respective policy is studied and analyzed through the lenses of the Bellman expectation (or optimality) which describes the evolution of values or rewards over time (cf. Kakade (2003); Sutton & Barto (2018); Kaelbling et al. (1996)). This is done by computing a *value function* which maps states to values and is updated online as new information becomes available.

However, value functions capture state information exclusively through its scalar value, which is a downside of model-free algorithms (see Sun et al. (2019) for theoretical comparisons between model-based and model-free approaches). Further, cumulative reward based trajectory (or policy) optimization is suboptimal for tasks such as path tracking where waypoints are unavailable, and for other problems that require the entire trajectory information to obtain an optimal control strategy. In particular, path tracking is the main focus of this work. Path tracking has been a central problem for autonomous vehicles (e.g., Schwarting et al. (2018); Aguiar & Hespanha (2007)), imitation learning, learning from demonstrations (cf. Hussein et al. (2017); Argall et al. (2009)), character animations (with mocap systems; e.g., Peng et al. (2018)), robot manipulation for executing plans returned by a solver (cf. Kaelbling & Lozano-Pérez (2011); Garrett et al. (2021)), and for flying drones (e.g., Zhou & Schwager (2014)), just to name a few. Typically, those problems are dealt with by using reference dynamics or are formulated as a control problem with a sequence of goals to follow.

In this work, we instead adopt a rough-path theoretical approach; specifically, we exploit path signatures (cf. Chevyrev & Kormilitzin (2016); Lyons (1998)), which have been widely studied as a useful geometrical feature representation of path, and have recently attracted the attention of the machine learning community (e.g., Chevyrev & Kormilitzin (2016); Kidger et al. (2019); Salvi et al. (2021); Morrill et al. (2021); Levin et al. (2013); Fermanian (2021)). By eReview and performance

evaluation of path trackinglishing the connection of algebra of path signatures to value functions, we tackle control problems for trajectory following in a novel manner.

Our framework predicated on signatures, named signature control, describes an evolution of signatures over a certain path. By demonstrating how it reduces to the Bellman equation as a special case, we show that the *S-function* representing the signatures of future path (we call it *path-to-go* in this paper) is cast as an effective generalization of value function. In addition, since an $S$-function naturally encodes information of a long trajectory, it is robust against misspecification of dynamics. Our signature control inherits some of the properties of signatures, namely, time-parameterization invariance, shift invariance, and tree-like equivalence (cf. Lyons et al. (2007); Boedihardjo et al. (2016)); as such, when applied to tracking problems, there is no need to specify waypoints.

In order to devise new algorithms from this framework, including model predictive controls (MPCs) (Camacho & Alba, 2013), we present path cost designs and their properties. In fact, our signature control generalizes the classical integral control (see Khalil (2002)); it hence shows robustness against unknown disturbances, which is demonstrated in robotic manipulator simulation.

**Our contributions:** We devise a novel framework based on path signatures for control problems named signature control. We define *Chen equation* for decision making and show how it reduces to the Bellman equation as a special instance, and discuss the relation to classical integral control. Our work offers a general approach that systematically lifts (generalizes) the classical Bellman-based dynamic programming to the space of paths; without the need of problem specific ad-hoc modifications, giving us more flexibility of devising algorithms on top. In addition, we present MPC algorithms accompanied by practical path cost designs and their properties. signature is a nice mathematical tool that has suitable properties for representing paths and has an algebraic property that is also well-fitted to dynamic programming based decision making. Finally, we analyze some of the advantages of our approach numerically and present several control and robotics applications. To this end, we show several simple numerical examples showcasing the benefits and demonstrating the concepts that inherit mathematical properties of the path signature, including its algebraic and geometric features.

**Notation:** Throughout this paper, we let $\mathbb{R}$, $\mathbb{N}$, $\mathbb{R}_{\geq 0}$, and $\mathbb{Z}_{>0}$ be the set of the real numbers, the natural numbers ($\{0, 1, 2, \ldots\}$), the nonnegative real numbers, and the positive integers, respectively. Also, let $[T] := \{0, 1, \ldots, T-1\}$ for $T \in \mathbb{Z}_{>0}$. The floor and the ceiling of a real number $a$ is denoted by $\lfloor a \rfloor$ and $\lceil a \rceil$, respectively. Let $\mathbb{T}$ denote time for random dynamical systems, which is defined to be either $\mathbb{N}$ (discrete-time) or $\mathbb{R}_{\geq 0}$ (continuous-time).

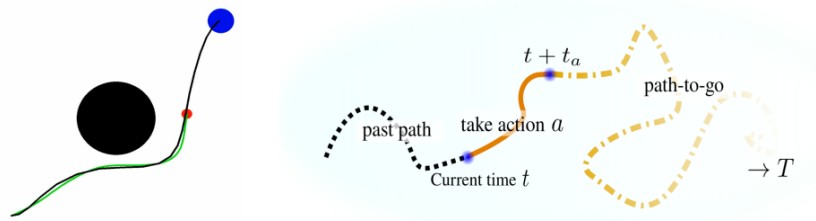

Figure 1: Left: simple tracking example. The black and blue circles represent an obstacle and the goal respectively. Given a feasible path (black line), a point-mass (red) follows this reference via minimization of deviation of signatures in an online fashion with optimized action repetitions. Right: illustration of path-to-go formulation as an analogy to value-to-go in the classical settings.

## 2 RELATED WORK

We introduce related lines of work on signature and its applications, controls/RLs, and path tracking.

**Path signature:** Path signatures are mathematical tools developed in rough path theoretical research (Lyons, 1998; Chen, 1954; Boedihardjo et al., 2016). For efficient computations of metrics over signatures, kernel methods (Hofmann et al., 2008; Aronszajn, 1950) are employed (Király & Oberhauser, 2019; Salvi et al., 2021; Cass et al., 2021; Salvi et al., 2021). Signatures have been widely applied to various applications, such as sound compression (Lyons & Sidorova, 2005), time series data analysis and regression (Gyurkó et al., 2013; Lyons, 2014; Levin et al., 2013), action and text recognitions (Yang et al., 2022; Xie et al., 2017), and neural rough differential equations (Morrill et al., 2021). Also, deep signature transform is proposed in (Kidger et al., 2019) with applications to reinforcement learning by embedding the signature transform as a layer in a recurrent neural network (RNN). The formulation of RL proposed by Kidger et al. (2019) is still within Bellman equation

based framework where states are augmented with signatures of past path that are updated using Chen's identity (Chen, 1954). In contrast, our framework is solely replacing this Bellman backup with signature based DP. More theory and practice for combining path signatures and machine learning are summarized in (Chevyrev & Kormilitzin, 2016; Fermanian, 2021).

**Control and RL:** Algorithm designs and sample complexity analyses for RL have been studied extensively (cf. (Kakade, 2003; Agarwal et al., 2019)). Value function based methods are widely adopted in RL for control problems. However model-free RL algorithms based on value function (Jiang et al., 2017; Haarnoja et al., 2018; Mnih et al., 2013; Wurman et al., 2022) treat the model as a black box and typically suffer from low sample efficiency. To alleviate this problem of value function based approaches, model-based RL methods learn one-step dynamics to exploit the dynamics across states (cf. (Sun et al., 2019; Du et al., 2021; Wang et al., 2019; Chua et al., 2018)). In practical algorithms, however, even small errors on one-step dynamics could diverge along multiple time steps, which hinders the performance (cf. Moerland et al. (2023)). To improve sample complexity and generalizability of value function based methods, on the other hand, successor features (e.g., Barreto et al. (2017)) have been developed for encoding representations of values for a diverse set of reward signals spanned by the features. We tackle the issue of value function based methods from another angle by capturing sufficiently rich information in a form of *path-to-go*, which is robust against long horizon problems; at the same time, it subsumes value function (and successor feature) updates. Our approach shares a similar idea proposed by Ohnishi et al. (2021) which controls with costs over spectrums of the Koopman operator in addition to a cumulative cost with sample complexity analysis, but that approach is not utilizing DP.

**Path tracking:** Path planning and path tracking are core techniques for autonomous robot navigation and control tasks. Traditionally, the optimal tracking control is achieved by using reference dynamics or by assigning time-varying waypoints (Paden et al., 2016; Schwarting et al., 2018; Aguiar & Hespanha, 2007; Zhou & Schwager, 2014; Patle et al., 2019). In practice, MPC is usually applied for tracking time-varying waypoints and PID controls are often employed when optimal control dynamics can be computed offline (cf. Khalil (2002)). Some of the important path tracking methodologies were summarized in (Rokonuzzaman et al., 2021); namely, pure pursuit (Scharf et al., 1969), Stanley controller (Thrun et al., 2007), linear control such as the linear quadratic regulator after feedback linearisation (Khalil, 2002), Lyapunov's direct method, robust or adaptive control using simplified or partially known system models, and MPC. Those methodologies are highly sensitive to time step sizes and requires analytical (and simplified) system models, and misspecifications of dynamics may also cause significant errors in tracking accuracy even when using MPC. Due to those limitations, many ad-hoc heuristics are often required when applying them in practice. Our signature based method applied to path tracking problems can systematically remedy some of those drawbacks in its vanilla form.

## 3 PRELIMINARIES

In this section, we introduce path signatures, their main properties, and present the problem setups.

### 3.1 PATH SIGNATURE

Let $\mathcal{X} \subset \mathbb{R}^d$ be a state space and suppose a path is defined over a compact time interval $[s,t]$ for $0 \leq s < t$ to be an element of $\mathcal{X}^{[s,t]}$; i.e., a continuous stream of states. The path signature is a collection of infinitely many features (scalar coefficients) of a path with *depth* one to infinite. Coefficients of each depth roughly correspond to the geometric characteristics of paths, e.g., displacement and area surrounded by the path can be expressed by coefficients of depth one and two.

The formal definition of path signatures is given below. We use $T((\mathcal{X}))$ to denote the space of formal power series and $\otimes$ to denote the tensor product operation (see Appendix A).

**Definition 3.1** (Path signatures (Lyons et al., 2007)). Let $\Sigma \subset \mathcal{X}^{[0,T]}$ be a certain space of paths. Define a map on $\Sigma$ over $T((\mathcal{X}))$ by $S(\sigma) \in T((\mathcal{X}))$ for a path $\sigma \in \Sigma$ where its coefficient corresponding to the basis $e_i \otimes e_j \otimes \ldots \otimes e_k$ is given by

$$S(\sigma)_{i,j,\ldots,k} := \int_{0 < \tau_k < T} \int_{0 < \tau_j < \tau_k} \ldots \int_{0 < \tau_i < \tau_j} dx_i \ldots dx_j dx_k.$$

The space $\Sigma$ is chosen so that the path signature $S(\sigma)$ of $\sigma \in \Sigma$ is well-defined.

Given a positive integer $m$, the truncated signature $S_m(\sigma)$ is defined accordingly by a truncation of $S(\sigma)$ (as an element of the quotient denoted by $T^m(\mathcal{X})$); see Appendix A for the definition of tensor algebra and related notations).

**Properties of the path signatures:** The basic properties of path signature allow us to develop our decision making framework and its extension to control algorithms. Such properties are also inherited by the algorithms we devise, providing several advantages over classical methods for tasks such as path tracking (further details are in Section 5 and 6). We summarize these properties below:

- The signature of a path is the same as the path shifted by a constant (shift invariance) and different time reparametrization (time parameterization invariance). Straightforward applications of signatures thus represent *shape* information of a path without the need to specify waypoints and/or absolute initial positions.
- A path is uniquely recovered from its signature up to tree-like equivalence (e.g., path with *detours*) and the magnitudes of coefficients decay as fast as depth increases. As such, (truncated) path signatures contain sufficiently rich information about the state trajectory, providing a valuable and compact representation of a path in several control problems.
- Any real-valued continuous (and nonlinear) map on the certain space of paths can be approximated to arbitrary accuracy by a linear map on the space of signatures (Arribas, 2018). This universality property enables us to construct a universal kernel which is used to compute the metric or cost of a generated path in our algorithms.
- The path signature has a useful algebraic property known as Chen's identity (Chen, 1954). It states that the signature of the concatenation of two paths can be computed by the tensor product of the signatures of the paths. Let $X : [a, b] \to \mathbb{R}^d$ and $Y : [b, c] \to \mathbb{R}^d$ be two paths. Then, it follows that
$$S(X * Y)_{a,c} = S(X)_{a,b} \otimes S(Y)_{b,c},$$
  where $*$ denotes the concatenation operation.

From the properties above, we can define a kernel operating over the space of trajectories. This will be critical to derive our control framework in section 4.

**Definition 3.2** (Signature Kernel). Let $X$ and $Y$ be two trajectories defined in two compact intervals $[a, a']$ and $[b, b']$. The signature kernel $K : \Sigma \times \Sigma \to \mathbb{R}$ is defined by
$$K(X, Y) := \langle S(X), S(Y) \rangle,$$
where the inner product $\langle \cdot, \cdot \rangle$ is the inner product defined for tensor algebra (see Appendix B for detailed definition). Efficient computation of the kernel was studied in (Salvi et al., 2021) or in (Király & Oberhauser, 2019) for truncated signatures.

We present several theoretical and numerical examples showcasing how these properties benefit control applications in Appendix G and H.

## 3.2 DYNAMICAL SYSTEMS AND PATH TRACKING

Since we are interested in cost definitions over the entire path and not limited to the form of the cumulative cost over a trajectory with fixed time interval (or integral along continuous time), Markov Decision Processes (MDPs; Bellman (1957)) are no longer the most suitable representation for the problem. Instead, we assume that the system dynamics of an agent is described by a stochastic dynamical system $\Phi$ (SDS; in other fields the term is also known as Random Dynamical System (RDS) (Arnold, 1995)). We defer the mathematical definition to Appendix C. In particular, let $\pi$ be a policy in a space $\Pi$ which defines the SDS $\Phi_\pi$ (it does not have to be a map from state space to action value). Roughly speaking, a SDS consists of two models:

- A model of the *noise*;
- A function representing the *physical* dynamics of the system.

Examples of stochastic dynamical systems include Markov chains, stochastic differential equations, and additive-noise systems, i.e.,
$$x_{t+1} = f(x_t) + \eta_t, \ \ x_0 \in \mathbb{R}^d, \ \ t \in [T],$$
where $f : \mathbb{R}^d \to \mathbb{R}^d$ represents the dynamics, and $\eta_t \in \mathbb{R}^d$ is the zero-mean i.i.d. additive noise vector. Intuitively, dynamical systems with an invariant noise-generating mechanism could be described as a stochastic dynamical system by an appropriate translation. We also note that SDS subsumes many practical systems studied in the control community.

Although the framework we propose is general, the main problem of interest is path tracking which we formally define below:

**Definition 3.3** (Path tracking). Let $\Gamma : \Sigma \times \Sigma \to \mathbb{R}_{\geq 0}$ be a cost function on the product of the spaces of paths over the nonnegative real number, satisfying:

$$\forall \sigma \in \Sigma : \ \Gamma(\sigma, \sigma) = 0; \qquad \forall \sigma^* \in \Sigma, \ \forall \sigma \in \Sigma \ \text{s.t.} \ \sigma \not\equiv_\sigma \sigma^* : \ \Gamma(\sigma, \sigma^*) > 0,$$

where $\equiv_\sigma$ is any equivalence relation of path in $\Sigma$. Given a reference path $\sigma^* \in \Sigma$ and a cost, the goal of path tracking is to find a policy $\pi^* \in \Pi$ such that a path $\sigma_\pi$ generated by the SDS $\Phi_\pi$ satisfies

$$\pi^* \in \arg\min_{\pi \in \Pi} \Gamma(\sigma_\pi, \sigma^*).$$

With these definitions we can now present a novel control framework, namely, signature control.

# 4 SIGNATURE CONTROL

A SDS creates a discrete or continuous path $\mathbb{T} \cap [0, T] \to \mathcal{X}$. For an SDS operating in discrete-time (i.e., $\mathbb{T}$ is $\mathbb{N}$) up to time $T$, we first interpolate each pair of two discrete points to create a path $[0, T] \to \mathcal{X}$. One may transform the obtained path onto $\Sigma$ as appropriate (see Appendix C for detailed description and procedures).

## 4.1 PROBLEM FORMULATION

Our signature control problem is described as follows:

**Problem 4.1** (signature control). Let $T \in [0, \infty)$ be a time horizon. The signature control problem is defined as

$$\text{Find } \pi^* \text{ s.t. } \pi^* \in \arg\min_{\pi \in \Pi} c\left(\mathbb{E}_\Omega\left[S\left(\sigma_\pi\left(x_0, T, \omega\right)\right)\right]\right), \tag{4.1}$$

where $c : T((\mathcal{X})) \to \mathbb{R}_{\geq 0}$ is a cost function over the space of signatures. $\sigma_\pi(x_0, T, \omega)$ is the (transformed) path for the SDS $\Phi_\pi$ associated with a policy $\pi$, $x_0$ is the initial state and $\omega$ is the realization for the *noise* model.

This problem subsumes the Markov Decision Process as we shall see in Section 4.2. The given formulation covers both discrete-time and continuous-time cases through interpolation over time. To simplify notation, we omit the details of probability space (e.g., realization $\omega$ and sample space $\Omega$) in the rest of the paper with a slight sacrifice of rigor (see Appendix C for detailed descriptions). Given a reference path $\sigma^*$, when $\Gamma(\sigma, \sigma^*) = c(S(\sigma))$ and $\equiv_\sigma$ denotes tree-like equivalence, signature control becomes the path tracking Problem 3.3. To effectively solve it we exploit dynamic programming over signatures in the next section.

## 4.2 DYNAMIC PROGRAMMING OVER SIGNATURES

Before introducing dynamic programming over signatures, we present the notion of *path-to-go* using Chen's identity.

**Path-to-go:** Without loss of generality, consider a finite-time dynamical system. Let $a \in \mathcal{A}$ be an *action* which basically constrains the realizations of path up to time $t_a$ (actions studied in MDPs are constraining the one-step dynamics from a given state). Given $T \in \mathbb{T}$, *path-to-go*, or the future path generated by $\pi$, refers to $\mathcal{P}^\pi$ defined by

$$\mathcal{P}^\pi(x, t) = \sigma_\pi(x, T - t), \ \forall t \in [0, T].$$

Under Markov assumption (see Appendix D), it follows that each realization of the path constrained by an action $a$ can be written as

$$\mathcal{P}^\pi(x, t) = \mathcal{P}^\pi_a(x, t) * \mathcal{P}^\pi(x^+, t_a + t), \quad \mathcal{P}^\pi_a(x, t) := \sigma_\pi(x, \min\{T - t, t_a\}),$$

where $x^+$ is the state reached after $t_a$ from $x$. This is illustrated in Figure 1 Right. To express the above relation in the signature form, we exploit the Chen's identity, and define the *signature-to-go* function (or in short $S$-function) $\mathcal{S}^\pi$ given by

$$\mathcal{S}^\pi(a, x, t) := \mathbb{E}\left[S(\mathcal{P}^\pi_a(x, t))|a\right]. \tag{4.2}$$

Using the Chen's identity, the law of total expectation, the Markov assumption, the properties of tensor product and the path transformation, we obtain the update rule:

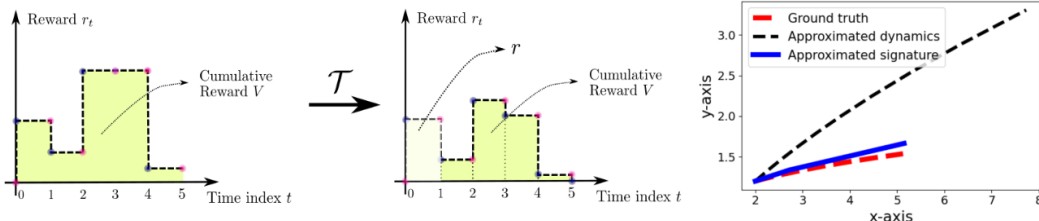

Figure 2: Left: illustrations of how a cumulative reward-to-go is represented by our path formulation. Right: an error of approximated one-step dynamics propagates through time steps; while an error on signature has less effect over long horizon.

**Theorem 4.2** (Signature Dynamic Programming for Decision Making). *Let the function $\mathcal{S}$ is defined by* (4.2). *Under the Markov assumption, it follows that*

$$\mathcal{S}^\pi(a, x, t) = \mathbb{E}\left[S(\mathcal{P}_a^\pi(x, t)) \otimes \mathcal{E}\mathcal{S}^\pi(x^+, t + t_a)|a\right]$$

*where the* expected $S$-function $\mathcal{E}\mathcal{S}^\pi$ *is defined by*

$$\mathcal{E}\mathcal{S}^\pi(x, t) := \mathbb{E}\left[\mathcal{S}^\pi(a, x, t)\right],$$

*where the expectation is taken over actions.*

**Truncated signature formulation:** For the $m$th-depth truncated signature (note that $m = \infty$ for signature with no truncation), we obtain,

$$(S(X) \otimes S(Y))_m = (S(X)_m \otimes S(Y)_m)_m =: S(X)_m \otimes_m S(Y)_m. \tag{4.3}$$

Therefore, when the cost only depends on the first $m$th-depth signatures, keeping track of the first $m$th-depth $S$-function $\mathcal{S}_m^\pi(a, x, t)$ suffices.

Following from these results, the cost function $c$ can be efficiently computed as

$$c(\mathcal{S}_m^\pi(a, x, t)) = c\left(\mathbb{E}\left[S_m(\mathcal{P}_a^\pi(x, t)) \otimes_m \mathcal{E}\mathcal{S}_m^\pi(x^+, t + t_a)|a\right]\right).$$

**Reduction to the Bellman equation:** Here we demonstrate that the formulation above subsumes the classical Bellman equation. Recall that the Bellman expectation equation w.r.t. action-value function or $Q$-function is given by

$$Q^\pi(a, x, t) = \mathbb{E}\left[r(a, x) + V^\pi(x^+, t + 1)|a\right], \tag{4.4}$$

where $V^\pi(x, t) = \gamma^t V^\pi(x, 0) = \mathbb{E}[Q^\pi(a, x, t)]$ for all $t \in \mathbb{N}$, where $\gamma \in (0, 1]$ is a discount factor. We briefly show how this equation can be described by a $S$-function formulation. Here, the action $a$ is the typical action input considered in MDPs. We suppose discrete-time system ($\mathbb{T} = \mathbb{N}$), and that the state is augmented by reward and time, and suppose $t_a = 1$ for all $a \in \mathcal{A}$. Let the depth of signatures to keep be $m = 2$. Then, by properly defining (see Appendix E for details) the interpolation and transformation, we obtain the path illustrated in Figure 2 Left over the time index and immediate reward. For this two dimensional path, note that a signature element of depth two represents the surface surrounded by the path (colored by yellow in the figure), which is equivalent to the value-to-go. As such, define the cost $c : T^2(\mathcal{X}) \to \mathbb{R}_{\geq 0}$ by $c(s) = -s_{1,2}$, and Chen equation becomes

$$c(\mathcal{S}_2^\pi(a, x, t)) = c\left(\mathbb{E}\left[S_2(\mathcal{P}_a^\pi(x, t)) \otimes_2 \mathcal{E}\mathcal{S}_2^\pi(x^+, t + 1)|a\right]\right)$$
$$= \mathbb{E}\left[-S_{1,2}(\mathcal{P}_a^\pi(x, t)) + c\left(\mathcal{E}\mathcal{S}_2^\pi(x^+, t + 1)\right)|a\right].$$

Now, since

$$c\left(\mathcal{S}_2^\pi(a, x, t)\right) = -Q^\pi(a, x, t), \ c\left(\mathcal{E}\mathcal{S}_2^\pi(x, t)\right) = -V^\pi(x, t), \ S_{1,2}(\mathcal{P}_a^\pi(x, t)) = r(a, x),$$

it reduces to the Bellman expectation equation (4.4).

Next, we present several practical applications that highlight some of the benefits of our signature control framework.

## 5 SIGNATURE MPC

First, we discuss an effective cost formulation over signatures to enable flexible and robust model predictive control (MPC). Then, we present additional numerical properties of path signatures that benefit signature control.

---

**Algorithm 1** Signature MPC

---

**Input**: initial state $x_0$; signature depth $m$; initial signature of past path $s_0 = \mathbf{1}$; # actions for rollout $N$; surrogate cost $\ell$ and regularizer $\ell_{\text{reg}}$; terminal $S$-function $\mathcal{TS}_m$; simulation model $\hat{\Phi}$

1: **while** not task finished **do**
2:     Observe the current state $x_t$
3:     Update the signature of past path: $s_t = s_{t-1} \otimes_m S(\sigma(x_{t-1}, x_t))$, where $S(\sigma(x_{t-1}, x_t))$ is the signature transform of the subpath traversed since the last update from $t - 1$ to $t$
4:     Compute the $N$ optimal future actions $\mathbf{a}^* := (a_0, a_1, \ldots, a_{N-1})$ using a simulation model $\hat{\Phi}$ that minimize the cost of the signature of the entire path (See Equation (5.1)).
5:     Run the first action $a_0$ for the associated duration $t_{a_0}$
6: **end while**

---

**Signature model predictive control:**   We present an application of Chen equation to MPC control– an iterative, finite-horizon optimization framework for control. In our signature model predictive control formulation, the optimization cost is defined over the signature of the full path being tracked i.e., the previous path seen so far and the future path generated by the optimized control inputs (e.g., distance from the reference path signature for path tracking problem). Our algorithm, given in Algorithm 1, works in the receding-horizon manner and computes a fixed number of actions (the execution time for the full path can vary as each action may have a different time scale; i.e., each action is taken effect up to optimized (or fixed) time $t_a \in \mathbb{T}$).

Given the signature $s_t$ of transformed past path (depth $m$) and the current state $x_t$ at time $t$, the actions are selected by minimizing a two-part objective which is the sum of the surrogate cost $\ell$ and some regularizer $\ell_{\text{reg}}$:

$$J = \begin{cases} \ell\left(s_t \quad \otimes_m \quad \mathbb{E}\left[S_m(\sigma_{\mathbf{a}}(x_t)) \otimes_m \mathcal{TS}_m(x_0, s_t, \sigma_{\mathbf{a}}(x_t))\right]\right) & \text{surrogate cost} \\ + \quad \ell_{\text{reg}}\left(\mathbb{E}\left[\mathcal{TS}_m(x_0, s_t, \sigma_{\mathbf{a}}(x_t))\right]\right) & \text{regularizer} \end{cases} \tag{5.1}$$

where the optimization variable $\mathbf{a} := (a_0, a_1, \ldots a_{N-1})$ is the sequence of actions, the path traced by an action sequence is $\sigma_{\mathbf{a}}(x_t)$, and $\mathcal{TS}_m : \mathcal{X} \times T^m(\mathcal{X}) \times \Sigma \to T^m(\mathcal{X})$ is the *terminal $S$-function* that may be defined arbitrarily (as an analogy to terminal value used in typical Bellman equation based MPC; see Appendix I for a comparison of a few options.).

Terminal $S$-function returns the signature of the terminal path-to-go. For the tracking problems studied in this work, we define the terminal subpath (path-to-go) as the final portion of the reference path starting from the closest point to the end-point of roll-out. This choice optimizes for actions up until the horizon anticipating that the reference path can be tracked afterward. We observed that this choice worked the best for simple classical examples analyzed in this work.

**Error explosion along time steps:**   We consider robustness against misspecification of dynamics. Figure 2 Right shows an example where the dashed red line is the ground truth trajectory with the true dynamics. When there is an approximation error on the one-step dynamics being modelled, the trajectory deviates significantly (black dashed line). On the other hand, when the same amount of error is added to each term of signature, the recovered path (blue solid line) is less erroneous. This is because signatures capture the entire (future) trajectory globally (see Appendix H.2 for details).

**Computations of signatures:**   We compute the signatures through the kernel computations (see Definition 3.2) using an approach in (Salvi et al., 2021). We emphasize that the discrete points we use for computing the signatures of (past/future) trajectories are not regarded as waypoints, and their placement has negligible effects on the signatures as long as they sufficiently maintain the "shape" of the trajectories; while the designs of waypoints for the classical Bellman-based MPC are critical.

## 6   Experimental Results

We conduct experiments on both simple classical examples and simulated robotic tasks. We also present a specific instance of signature control to the classical integral control to show its robustness against disturbance. For more experiment details, see Appendix J.

**Simple point-mass:**   We use double-integrator point-mass as a simple example to demonstrate our approach (as shown in Figure 1 Left). In this task, the velocity of a point-mass is controlled to reach

Table 1: Selected results on path following with an ant robot model. Comparing signature control, and baseline MPC and SAC RL with equally assigned waypoints. "Slow" means it uses more time steps than our signature control for reaching the goal.

| | Deviation (distance) from reference | | | |
| | Mean ($10^{-2}$m) | Variance ($10^{-2}$m) | # waypoints | reaching goal |
|---|---|---|---|---|
| signature control | **21.266** | **6.568** | N/A | success |
| baseline MPC | 102.877 | 182.988 | 880 | fail |
| SAC RL | 446.242 | 281.412 | 880 | fail |
| baseline MPC (slow) | 10.718 | 5.616 | 1500 | success |
| baseline MPC (slower) | 1.866 | 0.026 | 2500 | success |

a goal position while avoiding the obstacle in the scene. We first generate a collision-free path via RRT* (Karaman & Frazzoli, 2011) which can be suboptimal in terms of tracking speed. We then employ our Signature MPC to follow this reference path by producing the actions (*i.e.* velocities). Thanks to the properties of signature control, Signature MPC is able to optimize the tracking speed while matching the trajectory shape in the meantime. As a result, the solution produced by Signature MPC taking around 30 seconds while the reference path generated by RRT* path takes 72 seconds.

**Two-mass, spring, damper system:** To view the integral control (Khalil, 2002) within the scope of our proposed signature control formulation, recall a second depth signature term corresponding to the surface surrounded by the time axis, each of the state dimension, and the path, represents each dimension of the integrated error. In addition, a first depth signature term together with the initial state $x_0$ represent the immediate error, and the cost $c$ may be a weighted sum of these two. To test this, we consider two-mass, spring, damper system; the disturbance is assumed zero for planning, but is $0.03$ for executions. We compare Signature MPC where the cost is the squared Euclidean distance between the signatures of the reference and the generated paths with truncations upto the first and the second depth. The results of position evolutions of the two masses are plotted in Figure 3 Top. As expected, the black line (first depth) does not converge to zero error state while the blue line (second depth) does (see Appendix J for details). If we further include other signature terms, signature control effectively becomes a generalization of integral controls, which we will see for robotic arm experiments later.

**Ant path tracking:** In this task, an Ant robot is controlled to follow a "2"-shape reference path. The action optimized is the torque applied to each joint actuator. We test the tracking performances of signature control and baseline standard MPC on this problem. Also, we run soft actor-critic (SAC) (Haarnoja et al., 2018) RL algorithm where the state is augmented with time index to manage waypoints and the reward (negative cost) is the same as that of the baseline MPC. For the baseline MPC and SAC, we equally distribute 880 waypoints to be tracked along the path and the time stamp of each waypoint is determined by also equally dividing the total tracking time achieved by signature control. Table 1 compares the mean/variance of deviation (measured by distance in meter) from the closest of 2000 points over the reference path, and Figure 3 (Bottom Left) shows the resulting behaviors of MPCs, showing significant advantages of our method in terms of tracking accuracy. The performance of SAC RL is insufficient under the settings; this is expected given that we have no access to sophisticated waypoints over joints (see Peng et al. (2018) for the discussion). When more time steps are used (i.e., slower tracking), baseline MPC becomes a bit better. Note our method can also tune the trade-off between accuracy and progress through weight on regularizer.

**Robotic manipulator path tracking:** In this task, a robotic manipulator is controlled to track an end-effector reference path. Similar to the Ant task, 270 waypoints are equally sampled along the reference path for the baseline MPC method and SAC RL to track. To show robustness of signature control against unknown disturbance (torque: $N \cdot m$), we test different scales of disturbance force applied to each joint of the arm. The means/variances of the tracking deviation of the three approaches for selected cases are reported in Table 2 and the tracking paths are visualized in Figure 3 (Bottom Right). For all of the cases, our signature control outperforms the baseline MPC and SAC RL, especially when the disturbance becomes larger, the difference becomes much clearer. This is because the signature MPC is insensitive to waypoint designs but rather depends on the "distance" between the target path and the rollout path in the signature space, making the tracking speed adaptive.

Table 2: Results on path tracking with a robotic manipulator end-effector. Comparing signature control, and baseline MPC and SAC RL with equally assigned waypoints under unknown fixed disturbance.

| | | Deviation (distance) from reference | |
|---|---|---|---|
| | Disturbance $(\text{N} \cdot \text{m})$ | Mean $(10^{-2}\text{m})$ | Variance $(10^{-2}\text{m})$ |
| signature control | $+30$ | **1.674** | **0.002** |
| | $\pm 0$ | **0.458** | **0.001** |
| | $-30$ | **1.255** | **0.002** |
| baseline MPC | $+30$ | 2.648 | 0.015 |
| | $\pm 0$ | 0.612 | 0.007 |
| | $-30$ | 5.803 | 0.209 |
| SAC RL | $+30$ | 15.669 | 0.405 |
| | $\pm 0$ | 3.853 | 0.052 |
| | $-30$ | 16.019 | 0.743 |

Figure 3: Top (two-mass spring, damper system): the plots show the evolutions of positions of two masses for Signature MPC with/without second depth signature terms, showing how Signature MPC reduces to integral control. Down: (Left two; Ant) tracking behaviors of signature control (left) and baseline MPC (right) for the same reaching time, where green lines are the executed trajectories. (Right two; Robotic arm): tracking behaviors of signature control (left) and baseline MPC (right) under disturbance $-30$.

# 7 DISCUSSIONS

This work presented signature control, a novel framework that generalizes dynamic programming to reason over entire paths, instead of single timesteps. This allowed us to develop novel control algorithms such as Signature Model Predictive Control (Signature MPC), and apply them to path tracking problems. Since our framework is based on the signature transform, our algorithms inherit several useful features from it including time-parameterization invariance and shift invariance. We applied Signature MPC to path tracking in several domains, including a 2D point mass environment, a multi-joint ant, and a robotic manipulator, and showed superior tracking capability against baselines. We further demonstrated that our Signature MPC method is robust against misspecification of dynamics and significant disturbances. There are many promising avenues for future work (which is relevant to the current limitations of our work), such as developing a more complete theoretical understanding of guarantees provided by the signature control framework, leveraging hardware parallelization to accelerate Signature MPC and make it real-time, and developing additional reinforcement learning algorithms that inherit the benefits of our signature control framework. While we emphasize that the run times of MPC algorithms we used in this work for signature control and baseline are almost the same, adopting some of the state-of-the-art MPC algorithm running in real-time to our signature MPC is an important future work.

## 8    ETHICS STATEMENT

Our work is a fundamental research which potentially influences future control/RL/robotics work; as such, when misused (or carelessly used) for the military, self-driving cars, and robotics system interacting with humans, we mention that our work may negatively impact us.

## 9    REPRODUCIBILITY STATEMENT

We listed all the relevant parameters used in the experiments in Appendix J.

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

CONTENTS

## A  TENSOR ALGEBRA

We present the definition of tensor algebra here. In the main text, we used some of the notations, including $T((\mathcal{X}))$, defined below.

**Definition A.1** (Tensor algebra). Let $\mathcal{X}$ be a Banach space. The spaces of formal power series over $\mathcal{X}$ is defined by

$$T((\mathcal{X})) := \prod_{k=0}^{\infty} \mathcal{X}^{\otimes k},$$

where $\mathcal{X}^{\otimes k}$ is the tensor product of $k$ vector spaces ($\mathcal{X}$s). For $A = (a_0, a_1, \ldots)$, $B = (b_0, b_1, \ldots) \in T((\mathcal{X}))$, the addition $+$ and multiplication $\otimes$ are defined by

$$A + B = (a_0 + b_0, a_1 + b_1, \ldots), \ A \otimes B = (c_0, c_1, \ldots), \ c_k = \sum_{\ell=0}^{k} a_\ell \otimes b_{k-\ell}.$$

Also, $\lambda A = (\lambda a_0, \lambda a_1, \ldots)$ for any $\lambda \in \mathbb{R}$. The truncated tensor algebra for a positive integer $m$ is defined by the quotient $T^m(\mathcal{X})$

$$T^m(\mathcal{X}) := T((\mathcal{X}))/T_m,$$

where

$$T_m = \{A = (a_0, a_1, \ldots) \in T((\mathcal{X})) | a_0 = a_1 = \ldots = a_m = 0\}.$$

The equation (4.3) is immediate from the definition of the multiplication $\otimes$ of the formal power series and the signatures of $X \otimes Y$ upto depth $m$ only depend on the signatures of $X$ and $Y$ upto depth $m$.

## B  SIGNATURE KERNEL

We used signature kernels for computing the metric (or cost) for MPC problems. A path signature is a collection of infinitely many real values, and in general, the computations of inner product of a pair of signatures in the space of formal polynomials are intractable. Although, it is still not the exact computation in general, (Salvi et al., 2021) utilized a Goursat PDE to efficiently and approximately compute the inner product. The signature kernel is given below:

**Definition B.1** (Signature kernel (Salvi et al., 2021)). Let $\mathcal{X}$ be a $d$-dimensional space with canonical basis $\{e_1, \ldots, e_d\}$ equipped with an inner product $\langle \cdot, \cdot \rangle_{\mathcal{X}}$. Let $T(\mathcal{X}) := \bigoplus_{k=0}^{\infty} \mathcal{X}^{\otimes k}$ be the space of formal polynomials endowed with the same operators $+$ and $\otimes$ as $T((\mathcal{X}))$, and with the inner product

$$\langle A, B \rangle := \sum_{k=0}^{\infty} \langle a_k, b_k \rangle_{\mathcal{X}^{\otimes k}},$$

where $\langle \cdot, \cdot \rangle_{\mathcal{X}^{\otimes k}}$ is defined on basis elements $\{e_{i_1} \otimes \ldots \otimes e_{i_k} : \{i_1, \ldots, i_k\} \in \{1, \ldots, d\}^k\}$ as

$$\langle e_{i_1} \otimes \ldots \otimes e_{i_k}, e_{j_1} \otimes \ldots \otimes e_{j_k} \rangle_{\mathcal{X}^{\otimes k}} = \langle e_{i_1}, e_{j_1} \rangle_{\mathcal{X}} \ldots \langle e_{i_k}, e_{j_k} \rangle_{\mathcal{X}}.$$

Let $\overline{T(\mathcal{X})}$ be a completion of $T(\mathcal{X})$, and $\mathcal{H} := (\overline{T(\mathcal{X})}, \langle \cdot, \cdot \rangle)$ is a Hilbert space. The signature kernel $K : \Sigma \times \Sigma \to \mathbb{R}$ is defined by

$$K(X, Y) := \langle S(X), S(Y) \rangle,$$

for $X$ and $Y$ such that $S(X), S(Y) \in \overline{T(\mathcal{X})}$.

The overall properties of path signatures mentioned in Section 3.1 are illustrated in Figure 4.

## C  DETAILED PROBLEM SETTINGS

Here, we present the problem settings based on RDSs more carefully. First, we define the RDSs mathematically.

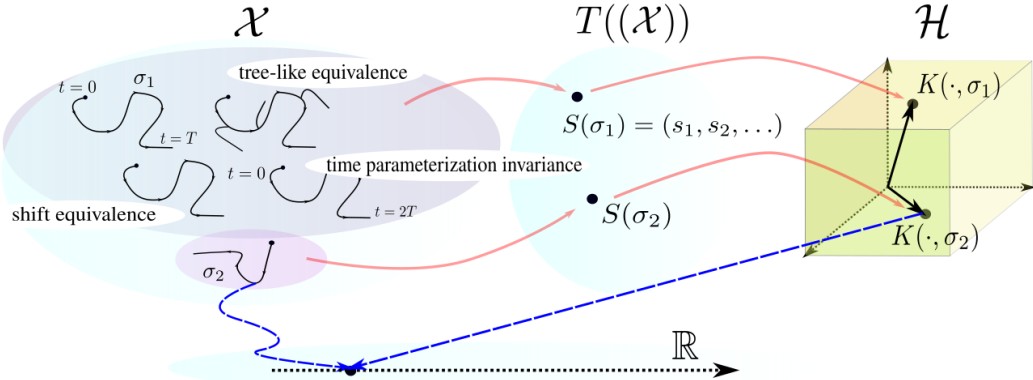

Figure 4: Illustrations of properties of path signatures. Paths in the space $\mathcal{X} \subset \mathbb{R}^d$ are uniquely transformed into signatures upto tree-like equivalence. One can construct an RKHS where the kernel represents the inner product between two signatures. This kernel is a universal kernel.

**Definition C.1** (Random dynamical systems (Arnold, 1995)). Let $(\Omega, P)$ be a probability space and $\{\theta_t\}_{t \in \mathbb{T}}$, where $\theta_t : \Omega \to \Omega$, $(\theta_t)_* P = P$, $\theta_0 = \mathrm{id}_\Omega$ and $\theta_s \circ \theta_t = \theta_{s+t}$ for all $s, t \in \mathbb{T}$, is a semi-group of measure preserving maps. Define a random dynamical system (RDS) by

$$\Phi : \mathbb{T} \times \Omega \times \mathbb{R}^d \to \mathbb{R}^d,$$

where

$$\Phi(0, \omega, x) = x, \quad \Phi(t + s, \omega, x) = \Phi(t, \theta_s(\omega), \Phi(s, \omega, x)), \quad \forall x \in \mathbb{R}^d.$$

An RDS is illustrated in Figure 5 as portrayed in Arnold (1995); Ghil et al. (2008).

In this work, in order to fully appreciate the generality of our framework, we view the policy $\pi \in \Pi$ as some *parameter* that defines an RDS. In particular, for simplicity, we assume that the random dynamical system generated by a policy $\pi \in \Pi$ shares the same sample space $\Omega$, and is denoted by $\Phi_\pi$. Roughly speaking, this means that the *noise mechanism* of RDSs is the same for all policies (not necessarily the same probability distribution). Also, the action $a \in \mathcal{A}$ is for constraining the event of downstream trajectories of RDS to be of some subset of $\Omega$, which we define $\Omega_a \subset \Omega$. Further, we suppose that $\bigsqcup_{a \in \mathcal{A}} \Omega_a = \Omega$ (see Appendix D for details).

**signature control:** We (re)define signature control carefully. Let $T \in \mathbb{T}$ be a time horizon, and define the map $\sigma_{\pi, F} : \mathcal{X} \times [0, T] \times \Omega \to \mathcal{X}^{[0,T]}$ by

$$
\begin{aligned}
&[\sigma_{\pi, F}(x_0, T, \omega)](t) \\
&= \begin{cases}
\Phi_\pi(t, \omega, x_0) & (\forall t \in \mathbb{T} \cap [0, T]) \\
[F(\Phi_\pi(\lfloor t \rfloor, \omega, x_0), \Phi_\pi(\lfloor t \rfloor + 1, \omega, x_0))](t - \lfloor t \rfloor) & (\forall t \in [0, T] \setminus \mathbb{T})
\end{cases},
\end{aligned}
$$

where $F : \mathcal{X} \times \mathcal{X} \to \mathcal{X}^{[0,1]}$ is an interpolation between two given points.

Also, let *practical* partition $\mathcal{D} = \{0 = t_0 < t_1 < \ldots < t_{k-1} = T\}$ of the time interval $[0, T]$ be such that there exists a sequence of actions $\{a_1, a_2, \ldots\}$ over that partition, i.e., $t_0, t_1, t_2...$ represent $0, t_{a_1}, t_{a_1} + t_{a_2},....$

Let $\mathcal{T} : \mathcal{X}^{[0,T]} \to \Sigma$ be some (possibly nonlinear) transformation such that, for any practical partition $\mathcal{D}$ of the time interval $[0, T]$, for any $j \in [k - 2]$, a pair of feasible paths $\sigma_1 \in \mathcal{X}^{[t_j, t_{j+1}]}, \sigma_2 \in \mathcal{X}^{[t_{j+1}, t_{j+2}]}$ satisfies

$$\sigma \equiv_\sigma \sigma_1 * \sigma_2 \implies \mathcal{T}(\sigma) \equiv_\sigma \mathcal{T}(\sigma_1) * \mathcal{T}(\sigma_2),$$

where $*$ denotes the concatenation of paths and $\equiv_\sigma$ is the tree-like equivalence relation. The interpolation $F$ and the transformation $\mathcal{T}$ are illustrated in Figure 6.

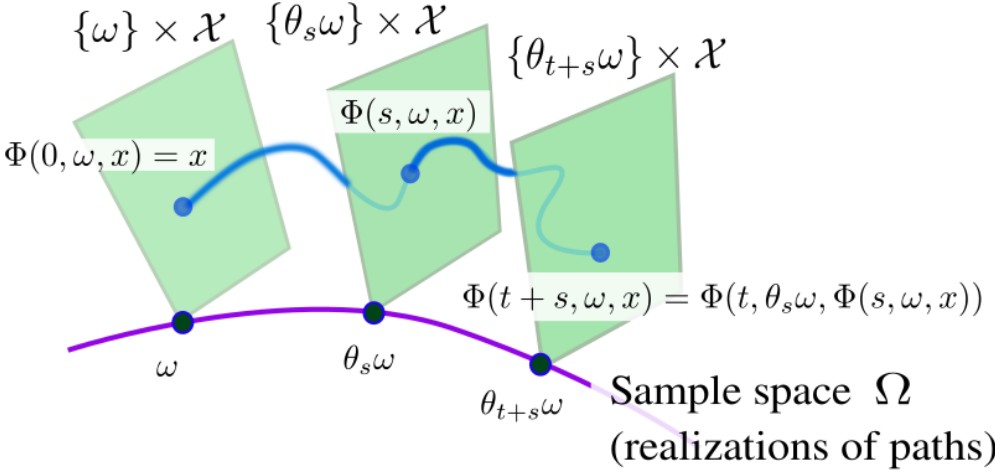

Figure 5: Random dynamical system consists of a model of the *noise* and the *physical* phase space. For each realization $\omega$, and initial state $x$, the RDS is the flow over sample space and phase space.

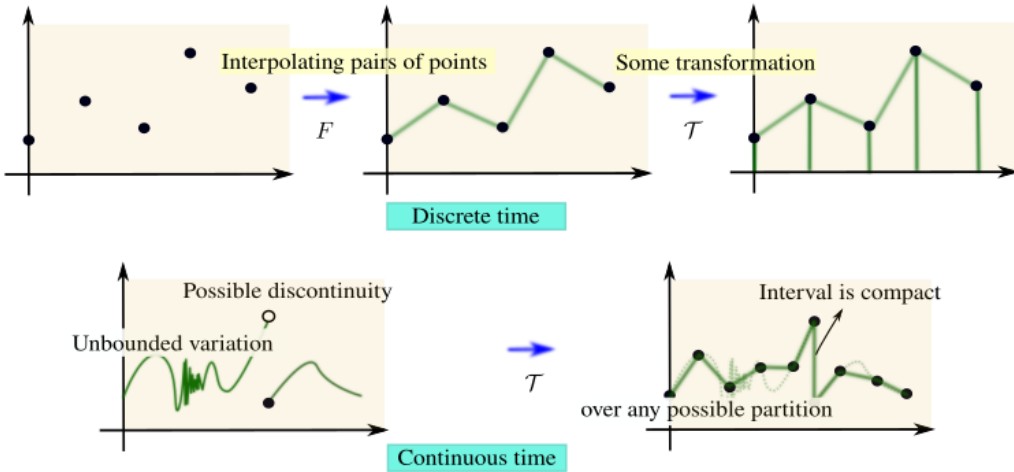

Figure 6: Top (discrete-time): interpolation for pairs of points will produce a path and one may possibly transform them. Down (continuous-time): a path generated by an RDS may include discontinuity or unbounded variation. Transformation makes it to be a path for which the signatures are defined. Discontinuous points may be interpolated (a path is defined over a compact interval), and unbounded variations may be overcome by down-sampling over points of any practical partition.

Then, the signature-based optimal control problem reads

$$\text{Find } \pi^* \text{ s.t. } \pi^* \in \underset{\pi \in \Pi}{\arg\min} \, c\left(\mathbb{E}_{\Omega}\left[S\left(\sigma_{\pi,F}^{\mathcal{T}}\left(x_0, T, \omega\right)\right)\right]\right),$$

where $c : T((\mathcal{X})) \to \mathbb{R}_{\geq 0}$ is a cost function on the space of formal power series, and $\sigma_{\pi,F}^{\mathcal{T}} := \mathcal{T} \circ \sigma_{\pi,F}$; we use, for simplicity, $\sigma_\pi$ instead of $\sigma_{\pi,F}^{\mathcal{T}}$ when $F$ and $\mathcal{T}$ are clear in the contexts.

## D  DETAILS OF PATH-TO-GO AND $S$-FUNCTION FORMULATIONS

Let the projection on $\mathcal{A} \times \mathcal{B}$ over $\mathcal{A}$ is denoted by $P_{\mathcal{A}} : (a, b) \mapsto a$. Without loss of generality, we assume that $\mathcal{X} = \mathcal{Y} \times \mathcal{O}$, and that $P_{\mathcal{O}}[\Phi_\pi(t, \omega, x)]$ is known when $P_{\mathcal{Y}}[x]$, $t$, and $\omega$ are given (i.e., $\mathcal{O}$

is the space of observations). Given $T \in \mathbb{T}$, define the *path-to-go* function $\mathcal{P}^\pi$ on $\mathcal{Y} \times \mathbb{T}$ over $\Sigma^\Omega$ by

$$\mathcal{P}^\pi(y,t)(\omega) = \sigma_\pi(x, T - t, \omega), \ \forall t \in [0, T].$$

Formal definition of Markov property used in this work is given below.

**Assumption 1** (Markov property (Arnold, 1995))**.** *For each action $a \in \mathcal{A}$, there exists $t_a \geq 0$ such that the RDS $\Phi_\pi$ satisfies the Markov property, i.e., for each $B \in 2^\mathcal{X}$, $a \in \mathcal{A}$, and $s \geq 0$,*

$$\Pr\left[\Phi(t_a + s, \omega, z) \in B | \Phi(t_a, \omega, z) = x, \omega \in \Omega_a\right] = \Pr\left[\omega | \Phi(s, \omega, x) \in B\right]. \tag{D.1}$$

**Remark D.1.** When $\mu(\Omega_a) = 0$, we can still assign the probability of the right hand side of (D.1) to its left hand side; however, one can define arbitrarily the probability of a future path conditioned on $\omega \in \Omega_a$ and it does not harm the current arguments for now.

Now, the path-to-go formulation is reexpressed by

$$\mathcal{P}^\pi(y,t)(\omega) = \mathcal{P}_a^\pi(y,t)(\omega) * \mathcal{P}^\pi(y^+, t + t_a)(\theta_{t_a}\omega),$$

where

$$\mathcal{P}_a^\pi(y,t)(\omega) := \sigma_\pi(x, \min\{T - t, t_a\}, \omega),$$
$$y^+ = P_\mathcal{Y}\Phi(t_a, \omega, x).$$

Then, Theorem 4.2 is proved as follows:

*Proof of Theorem 4.2.* Using the Chen's identity (first equality), tower rule (second equality), Assumption 1 (third equality), and the properties of tensor product and the transformation (first and third equalities) we obtain

$$\mathcal{S}^\pi(a,y,t)$$
$$= \mathbb{E}\left[S(\mathcal{P}_a^\pi(y,t)(\omega)) \otimes A | \omega \in \Omega_a\right] = \mathbb{E}\left[\mathbb{E}\left[S(\mathcal{P}_a^\pi(y,t)(\omega)) \otimes A | \mathcal{P}_a^\pi(y,t)(\omega), \omega \in \Omega_a\right] | \omega \in \Omega_a\right]$$
$$= \mathbb{E}\left[S(\mathcal{P}_a^\pi(y,t)(\omega)) \otimes \mathbb{E}\left[A | \mathcal{P}_a^\pi(y,t)(\omega)\right] | \omega \in \Omega_a\right]$$
$$= \mathbb{E}\left[S(\mathcal{P}_a^\pi(y,t)(\omega)) \otimes \mathcal{E}\mathcal{S}^\pi(y^+, t + t_a) | \omega \in \Omega_a\right]$$

where

$$A := S(\mathcal{P}^\pi(y^+, t + t_a)(\theta_{t_a}\omega)),$$

and the *expected S-function* $\mathcal{E}\mathcal{S}^\pi : \mathcal{Y} \times \mathbb{T} \to T((\mathcal{X}))$ is defined by

$$\mathcal{E}\mathcal{S}^\pi(y,t) := \mathbb{E}_\Omega\left[\mathcal{S}^\pi(b(\omega), y, t)\right],$$

and $b : \Omega \to \mathcal{A}$ is defined by $b(\omega) = a$ for $\omega \in \Omega_a$. $\qquad\square$

# E   DETAILS ON REDUCTION TO BELLMAN EQUATIONS

Here, we carefully show how Chen equation reduces to Bellman expectation equation:

$$Q^\pi(a,x,t) = \mathbb{E}_\Omega\left[r(a,x,\omega) + V^\pi(x^+, t + 1) \big| \omega \in \Omega_a\right], \tag{E.1}$$

We suppose $\mathcal{X} := \mathcal{Z} \times \mathbb{R}_{\geq 0} \times \mathbb{R}_{\geq 0} \subset \mathbb{R}^d$, for $d > 2$, is the state space augmented by the immediate reward and time, and suppose $t_a = 1$ for all $a \in \mathcal{A}$. Let $m = 2$. We define the interpolation $F$, the transformation $\mathcal{T}$, and the cost function $c$ so that

$$\forall x,y \in \mathcal{X} \text{ s.t. } x_{d-1:d} = [r_x, t_x], y_{d-1:d} = [r_y, t_x + 1]:$$

$$F(x,y)(\tau) = \begin{cases} [r_x + 2\tau \cdot (r_y - r_x), t_x] & (\tau \in [0, 0.5]) \\ [r_y, t_x + 2(\tau - 0.5)] & (\tau \in (0.5, 1]) \end{cases},$$

$$\forall \sigma : [s,t] \to \mathbb{R}_{\geq 0} \times \mathbb{R}_{\geq 0} \text{ s.t. } s, t \in \mathbb{N}, \ s < t:$$

$$\mathcal{T}(\sigma)(\tau + s) = \begin{cases} \left[2\tau\gamma^s[\sigma(s+1)]_{d-1}, s\right], & (\tau \in [0, 0.5]) \\ \left[\gamma^{\xi_1(\tau)}[\sigma(\tau+s)]_{d-1}, [\sigma(\tau+s)]_d\right], & (\tau \in [0.5, t - s - 0.5)) \\ \left[\gamma^{\lfloor\xi_2(\tau)\rfloor}[\sigma(\xi_2(\tau))]_{d-1}, [\sigma(\xi_2(\tau))]_d\right], & (\tau \in [t - s - 0.5, t - s - 0.25)) \\ \left[4(t - s - \tau)\gamma^{t-1}[\sigma(t)]_{d-1}, [\sigma(t)]_d\right], & (\tau \in [t - s - 0.25, t - s]) \end{cases}$$

$$c(s) = -s_{1,2},$$

where

$$\xi_1(\tau) = \begin{cases} \max\left\{2(\tau - \lfloor\tau\rfloor) + \lfloor\tau\rfloor + s - 1, 0\right\}, & (\tau - \lfloor\tau\rfloor \leq 0.5) \\ \lfloor\tau\rfloor + s, & (\tau - \lfloor\tau\rfloor > 0.5), \end{cases}$$

and $\xi_2(\tau) = 2\tau - (t - s - 0.5) + s$.

Then Chen equation reduces to the Bellman equation (E.1) by

$$c(\mathcal{S}_2^\pi(a, y, t))$$
$$= c\left(\mathbb{E}\left[S_2(\mathcal{P}_a^\pi(y, t)(\omega)) \otimes_2 \mathcal{ES}_2^\pi(y^+, t+1) | \omega \in \Omega_a\right]\right)$$
$$= \mathbb{E}\left[-S_{1,2}(\mathcal{P}_a^\pi(y, t)(\omega)) + c\left(\mathcal{ES}_2^\pi(y^+, t+1)\right) | \omega \in \Omega_a\right]$$

Put

$$c\left(\mathcal{S}_2^\pi(a, y, t)\right) = -Q^\pi(a, y, t), \quad c\left(\mathcal{ES}_2^\pi(y, t)\right) = -V^\pi(y, t), \quad S_{1,2}(\mathcal{P}_a^\pi(y, t)(\omega)) = r(a, y, \omega),$$

and it reduces to Bellman expectation equation.

**Optimality:** Next, we briefly cover optimality; i.e., we present Chen optimality equation. Optimality is tricky for Chen formulation because some relation between policy and action is required in addition to the Markov assumption. To obtain our Chen optimality, we make the following assumption.

**Assumption 2** (Relations between policy and action). *For any policy $\pi \in \Pi$, state $x \in \mathcal{X}$, time $t \in \mathbb{T} \cap (0, T]$, and an action $a \in \mathcal{A}$, there exists a policy $\pi' \in \Pi$ such that*

$$\mathbb{E}_\Omega\left[\mathcal{S}^{\pi'}(b(\omega), P_{\mathcal{Y}}(x_0), 0)\right]$$
$$= \mathbb{E}_\Omega\left[\mathcal{S}^\pi(b(\omega), P_{\mathcal{Y}}(x_0), 0) \big| (\Phi_\pi(t, \omega, x_0) = x) \Longrightarrow (\theta_t\omega \in \Omega_a)\right].$$

*Also, there exists $a \in \mathcal{A}$ such that $\pi' = \pi$.*

Given a positive integer $m$ and a cost function $c : T^m(\mathcal{X}) \to \mathbb{R}_{\geq 0}$, suppose $\pi^*$ satisfies

$$c\left(\mathcal{ES}_m^{\pi^*}(P_{\mathcal{Y}}(x_0), 0)\right) = \inf_{\pi \in \Pi} c\left(\mathcal{ES}_m^\pi(P_{\mathcal{Y}}(x_0), 0)\right).$$

Then, under Assumption 2, Chen optimality reads

$$c\left(\mathbb{E}_\Omega\left[\mathcal{S}^{\pi^*}(b(\omega), P_{\mathcal{Y}}(x_0), 0)\right]\right)$$
$$= \min_{a \in \mathcal{A}} c\left(\mathbb{E}_\Omega\left[\mathcal{S}^{\pi^*}(b(\omega), P_{\mathcal{Y}}(x_0), 0) \big| (\Phi_{\pi^*}(t, \omega, x_0) = x) \Longrightarrow (\theta_t\omega \in \Omega_a)\right]\right), \quad \text{(E.2)}$$

when the right hand side is defined.

Therefore, with the same settings as the case of reduction to Bellman expectation equation, we have

$$c\left(\mathbb{E}_\Omega\left[\mathcal{S}^{\pi^*}(b(\omega), P_{\mathcal{Y}}(x_0), 0) \big| \sigma_{\pi^*}(x_0, t, \omega)(t) = x\right]\right)$$
$$= \min_{a \in \mathcal{A}} c\left(\mathbb{E}_\Omega\left[\mathcal{S}^{\pi^*}(b(\omega), P_{\mathcal{Y}}(x_0), 0) \big| (\sigma_{\pi^*}(x_0, t, \omega)(t) = x) \wedge (\theta_t\omega \in \Omega_a)\right]\right),$$

and we obtain

$$c\left(\mathbb{E}_\Omega\left[\mathcal{S}^{\pi^*}(b(\omega), P_{\mathcal{Y}}(x_0), 0) \big| \sigma_{\pi^*}(x_0, t, \omega)(t) = x\right]\right)$$
$$= c\left(\mathbb{E}_\Omega\left[S(\sigma_{\pi^*}(x_0, t, \omega)) \otimes \mathcal{S}^{\pi^*}(b(\theta_t\omega), P_{\mathcal{Y}}(x), t) \big| \sigma_{\pi^*}(x_0, t, \omega)(t) = x\right]\right)$$
$$= c\left(\mathbb{E}_\Omega\left[S(\sigma_{\pi^*}(x_0, t, \omega)) \big| \sigma_{\pi^*}(x_0, t, \omega)(t) = x\right]\right) + c\left(\mathcal{ES}_2^{\pi^*}(P_{\mathcal{Y}}(x), t)\right)$$
$$= c\left(\mathbb{E}_\Omega\left[S(\sigma_{\pi^*}(x_0, t, \omega)) \big| \sigma_{\pi^*}(x_0, t, \omega)(t) = x\right]\right) +$$
$$\min_{a \in \mathcal{A}} \mathbb{E}\left[-S_{1,2}(\mathcal{P}_a^{\pi^*}(P_{\mathcal{Y}}(x), t)(\omega)) + c\left(\mathcal{ES}_2^{\pi^*}(P_{\mathcal{Y}}(x^+), t+1)\right) | \omega \in \Omega_a\right],$$

from which it follows that

$$V^{\pi^*}(y, t) = \max_{a \in \mathcal{A}} \mathbb{E}\left[r(a, y, \omega) + V^{\pi^*}(y^+, t+1)\right].$$

# F  INFINITE TIME INTERVAL EXTENSION OF CHEN FORMULATION

Extending Chen equation to infinite time interval requires an argument of the extended real line. Let $[0, \infty] \subset \overline{\mathbb{R}}$ be the subset of the extended real line $\overline{\mathbb{R}}$. Now, we make the following assumption:

**Assumption 3.** *There exists a homeomorphism $\psi : [0, \infty] \to [0, T]$ for some $T > 0$ such that for any policy $\pi \in \Pi$, initial state $x_0 \in \mathcal{X}$, and realization $\omega \in \Omega$, the limit*

$$\lim_{\tau \to \infty} \sigma^{\mathcal{T}}_{\pi, F}(x_0, \tau, \omega)(\tau)$$

*exists and the path $\sigma^{\mathcal{T}}_{\pi, F}$ can be continuously extended to $[0, \infty]$. In addition, the path $\sigma_\pi : \mathcal{X} \times \Omega \to \Sigma$ defined by*

$$\sigma_\pi(x_0, \omega)(t) = \left[ \lim_{\tau \to \infty} \sigma^{\mathcal{T}}_{\pi, F}(x_0, \tau, \omega) \right] (\psi^{-1}(t)), \ \ \forall x_0 \in \mathcal{X}, \ \omega \in \Omega, \ t \in [0, T],$$

*is an element of $\Sigma$.*

Now, we redefine (for avoiding introducing more notations)

$$\mathcal{P}^\pi(y)(\omega) = \sigma_\pi(x, \omega),$$

and Chen equation becomes

$$
\begin{aligned}
&\mathcal{S}^\pi(a, y) \\
&= \mathbb{E}\left[ S(\mathcal{P}^\pi_a(y, 0)(\omega)) \otimes A | \omega \in \Omega_a \right] = \mathbb{E}\left[ \mathbb{E}\left[ S(\mathcal{P}^\pi_a(y, 0)(\omega)) \otimes A | \mathcal{P}^\pi_a(y, 0)(\omega), \omega \in \Omega_a \right] | \omega \in \Omega_a \right] \\
&= \mathbb{E}\left[ S(\mathcal{P}^\pi_a(y, 0)(\omega)) \otimes \mathbb{E}\left[ A | \mathcal{P}^\pi_a(y, 0)(\omega) \right] | \omega \in \Omega_a \right] \\
&= \mathbb{E}\left[ S(\mathcal{P}^\pi_a(y, 0)(\omega)) \otimes \mathcal{E}\mathcal{S}^\pi(y^+) | \omega \in \Omega_a \right]
\end{aligned}
$$

where $A$ is redefined by

$$A := S(\mathcal{P}^\pi(y^+)(\theta_{\psi(t_a)}\omega)).$$

To see how it reduces to infintie horizon Bellman expectation equation, note that one cannot consider time axis now because it diverges and signatures are no longer defined. Therefore, instead we consider $\mathcal{O}$ to be a space of discount factor and discounted cumulative reward, i.e., one dimension of $\mathcal{O}$ evolves as $1, \gamma, \gamma^2, \ldots$ and the other dimension is given by $r_0, r_0 + \gamma r_1, r_0 + \gamma r_1 + \gamma^2 r_2, \ldots$. Extracting the path over discounted cumulative reward, and transforming it so that it starts from $0$, the first depth signature (displacement) corresponds to the value-to-go. We omit the details but we mention that the value function is again captured by signatures.

# G  SEPARATIONS FROM CLASSICAL APPROACH

One may think that one can augment the state with signatures and give reward at the very end of the episode to encode the value over the entire trajectory within the classical Bellman based framework. There are obvious drawbacks for this approach; (1) for the infinite horizon case where the terminal state or time is unavailable, one cannot give any reward, and (2) input dimension for the value function becomes very large with signature augmentation. Here, in addition to the above, we show separations from the classical Bellman based approach from several point of views. Let $\mathcal{S}^\pi(a, y, t)$ $(\mathcal{E}\mathcal{S}(y, t))$ be the $S$-function (expected $S$-function) and $Q^\pi(a, y, s, t)$ $(V^\pi(y, s, t))$ be the $Q$-function (value function) where $s$ represents the signature of the past path.

## G.1  COST AND EXPECTATION ORDER

If the cost $c$ is linear (e.g., the case of reduction to Bellman equation), then the cost to be mimimized can be reformulated as

$$c\left( \mathbb{E}_\Omega \left[ S_m \left( \sigma^{\mathcal{T}}_{\pi, F}(y_0, T, \omega) \right) \right] \right) = \mathbb{E}_\Omega \left[ c\left( S_m \left( \sigma^{\mathcal{T}}_{\pi, F}(y_0, T, \omega) \right) \right) \right].$$

However, in general, the order is not exchangable. We saw that Chen equation reduces to Bellman equation and therefore for any MDP over the state augmented by signatures (and horizon $T$), it is easy to see that there exists an interpolation, a transpotation, and a cost $c$ such that

$$c\left( \mathcal{E}\mathcal{S}^\pi_m(y_0, 0) \right) = V^\pi(y_0, \mathbf{1}, 0).$$

On the other hand, the opposite does not hold in general.

**Claim G.1.** *There exist a 3-tuple $(\mathcal{X}, \mathcal{A}, P_a)$ where $P_a$ is the transition kernel for action $a \in \mathcal{A}$, a set of randomized policies ($\pi(a|x)$ is the probability of taking action $a \in \mathcal{A}$ at $x \in \mathcal{X}$ under the policy $\pi$) $\Pi$, an initial state $y_0$, and the cost function $c$ of signature control, such that there is no immediate reward function $r$ that satisfies*

$$\arg\min_{\pi \in \Pi} c\left(\mathcal{ES}_m^\pi(y_0, 0)\right) = \arg\min_{\pi \in \Pi} V^\pi(y_0, \mathbf{1}, 0).$$

*Proof.* Let $\mathcal{X} = \mathcal{Y} = \mathbb{R}$, $\mathcal{A} = \{a_{-1}, a_1\}$, $\mathbb{T} = \mathbb{N}$, $y_0 = 0$, $T = 1$ and

$$P_{a_{-1}}(0, -1) = 1, \ P_{a_1}(0, 1) = 1.$$

Also, let $\Pi = \{\pi_1, \pi_2, \pi_3\}$ where

$$\pi_1(a_1|0) = 1, \ \pi_2(a_{-1}|0) = 1, \ \pi_3(a_1|0) = 0.5, \ \pi_3(a_{-1}|0) = 0.5,$$

and let $c : T^1(\mathcal{X}) \to \mathbb{R}_{\geq 0}$ be

$$c(s) = |s_1|.$$

The optimal policy for signature control is then $\pi_3$, i.e.,

$$\{\pi_3\} = \arg\min_{\pi \in \Pi} c\left(\mathcal{ES}_m^\pi(y_0, 0)\right).$$

Now, because we have

$$V^{\pi_1}(y_0, \mathbf{1}, 0) = \mathbb{E}_{\Omega_{a_{-1}}}\left[r(a_{-1}, y_0, \omega)\right],$$
$$V^{\pi_2}(y_0, \mathbf{1}, 0) = \mathbb{E}_{\Omega_{a_1}}\left[r(a_1, y_0, \omega)\right],$$
$$V^{\pi_3}(y_0, \mathbf{1}, 0) = \frac{\mathbb{E}_{\Omega_{a_{-1}}}\left[r(a_{-1}, y_0, \omega)\right] + \mathbb{E}_{\Omega_{a_1}}\left[r(a_1, y_0, \omega)\right]}{2},$$

possible immediate reward to consider are only $\mathbb{E}_{\Omega_{a_{-1}}}\left[r(a_{-1}, y_0, \omega)\right]$ and $\mathbb{E}_{\Omega_{a_1}}\left[r(a_1, y_0, \omega)\right]$. It is straightforward to see that

$$\pi_3 \in \arg\min_{\pi \in \Pi} V^\pi(y_0, \mathbf{1}, 0)$$

only if

$$\mathbb{E}_{\Omega_{a_{-1}}}\left[r(a_{-1}, y_0, \omega)\right] = \mathbb{E}_{\Omega_{a_1}}\left[r(a_1, y_0, \omega)\right].$$

However, for any reward function satisfying this equation we obtain

$$\{\pi_3\} \neq \arg\min_{\pi \in \Pi} V^\pi(y_0, \mathbf{1}, 0).$$

$\square$

## G.2 SAMPLE COMPLEXITY

We considered randomized policies class above. What if the dynamics is deterministic ($\Omega$ is a singleton)? For deterministic finite horizon case, technically, the cost over path can be represented by both the cost function with $S$-function and $Q$-function. The difference is the *steps* or sample complexity required to find an optimal path. Because $S$-function captures strictly more information than $Q$-function, it should show sample efficiency in certain problems even for deterministic case. Here, in particular, we show that there exists a signature control problem which is more efficiently solved by the use of $S$-function than that of $Q$-function. (We do not discuss typical lower bound arguments of RL sample complexity; giving certain convergence guarantees with lower bound arguments is an important future work.)

To this end, we define Signature MDP:

**Definition G.1** (Finite horizon, time-dependent signature MDP)**.** Finite horizon, time-dependent signature MDP is the 8-tuple $(\mathcal{X}, \mathcal{A}, m, \{P\}_t, F, \{r\}_t, T, \mu)$ which consists of

- finite or infinite state space $\mathcal{X}$

- discrete or infinite action space $\mathcal{A}$

- signature depth $m \in \mathbb{Z}_{>0}$

- transition kernel $P_{a,t}$ on $\mathcal{X} \times \mathcal{X}$ for action $a \in \mathcal{A}$ and time $t \in [T]$

- signature is updated through concatenation of past path and the immediate path which is the interpolation of the current state and the next state by $F$

- reward $r_t$ which is a time-dependent mapping from $\mathcal{X} \times T^m(\mathcal{X}) \times \mathcal{A}$ to $\mathbb{R}$ for time $t \in [T]$

- positive integer $T \in \mathbb{Z}_{>0}$ defining time horizon

- initial state distribution $\mu$

Further, we call an algorithm $Q$-table ($\mathcal{S}$-table) based if it accesses state $x$ exclusively through $Q$-table ($\mathcal{S}$-table) for all $x \in \mathcal{X}$. Now, we obtain the following claim.

**Claim G.2.** *There exists a finite horizon, time-dependent signature MDP with a set of deterministic policies $\Pi$ and with a* known *reward $\{r\}_t$ such that the number of samples (trajectories) required in the worst case to determine an optimal policy is strictly larger for any $Q$-table based algorithm than a $\mathcal{S}$-table based algorithm.*

*Proof.* Let the first MDP $\mathcal{M}_1$ be given by $T = 3$, $\mathcal{X} = \mathcal{Y} := \{[0,0],[1,1],[2,2],[2,3],[-1,1],[0,1],[4,0]\} \subset \mathbb{R}^2$, $\mathcal{A} := \{a_1, a_2\}$, $m = 2$, $F$ is linear interpolation of any pair of points, $\mu([0,0]) = \Pr[y_0 = [0,0]] = 1$, and

$$P_{a_1,0}([0,0],[1,1]) = P_{a_1,1}([1,1],[2,2]) = P_{a_1,2}([2,2],[2,3]) =$$
$$= P_{a_2,0}([0,0],[-1,1]) = P_{a_2,1}([-1,1],[2,2]) = P_{a_2,2}([2,2],[4,0]) = 1$$

Also, let $\{r\}_t$ satisfy that

$$\forall t \in [T-1]: \ r_t = 0, \quad r_{T-1}(x,s,a) = |s_{1,2}^+|,$$

where $s^+$ is the signature of entire path that is deterministically obtained from state $x$ at time $T-1$, past path signature $s$, and action $a$ (note we do not know the transition but only the output $|s_{1,2}^+|$). The possible deterministic trajectories (or policies) of state-action pairs are the followings:

$$(([0,0],a_1),([1,1],a_1),([2,2],a_1),([2,3]))$$
$$(([0,0],a_1),([1,1],a_2),([2,2],a_1),([2,3]))$$
$$(([0,0],a_1),([1,1],a_1),([2,2],a_2),([4,0]))$$
$$(([0,0],a_1),([1,1],a_2),([2,2],a_2),([4,0]))$$
$$(([0,0],a_2),([-1,1],a_1),([2,2],a_1),([2,3]))$$
$$(([0,0],a_2),([-1,1],a_2),([2,2],a_1),([2,3]))$$
$$(([0,0],a_2),([-1,1],a_1),([2,2],a_2),([4,0]))$$
$$(([0,0],a_2),([-1,1],a_2),([2,2],a_2),([4,0]))\,.$$

The optimal trajectories are the last two. Let the second MDP $\mathcal{M}_2$ be the same as $\mathcal{M}_1$ except that

$$P_{a_2,2}([2,2],[2,3]) = 1.$$

Suppose we obtain $Q$-table for the first six trajectories of $\mathcal{M}_1$. At this point, we cannot distinguish $\mathcal{M}_1$ and $\mathcal{M}_2$ exlusively from the $Q$-table; hence at least one more trajectory sample is required to determine the optimal policy for any $Q$-table based algorithm. On the other hand, suppose we obtain $\mathcal{S}$-table for the first six trajectories of $\mathcal{M}_1$. Then, the $S$-table at state $x = [2,2]$ with depth $m \geq 1$ determines the transition at $x = [2,2]$ for both actions; hence, we know the cost of the the last two trajectories without executing it. $\square$

## H    OTHER NUMERICAL EXAMPLES: SANITY CHECK

In this section, we present several numerical examples backing the basic properties of Chen equation.

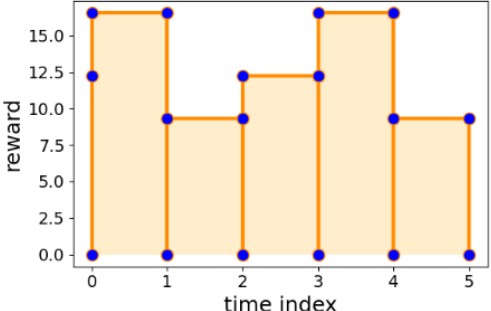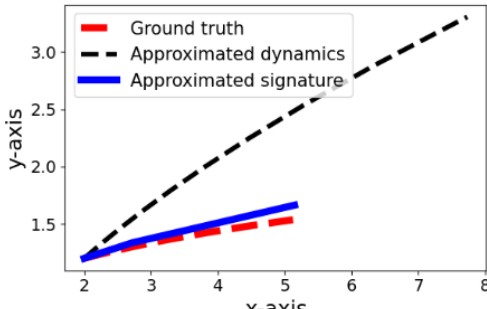

Figure 7: Left: A concatenated transformed path; under tree-like equivalence, the signature corresponding to this path is the same as that of the entire transformed path from time $0$ to $5$. As such, the area surrounded by such path represents the cumulative reward. Right: comparisons of (1) ground-truth path, (2) the path following approximated one-step dynamics, and (3) the reconstructed path from erroneous signatures.

Table 3: Observation vector for each state; rounded off to two decimal places.

| State | 1 | 2 | 3 | 4 | 5 | 6 | 7 | 8 | 9 | 10 |
|-------|------|-------|-------|-------|-------|------|-------|------|------|-------|
| $o_1$ | 1.92 | 4.38 | 7.80 | 2.76 | 9.58 | 3.58 | 6.83 | 3.70 | 5.03 | 7.73 |
| $o_2$ | 6.22 | 7.85 | 2.73 | 8.02 | 8.76 | 5.01 | 7.13 | 5.61 | 0.14 | 8.83 |
| $o_1 + o_2$ | 8.14 | 12.23 | 10.53 | 10.78 | 18.34 | 8.59 | 13.96 | 9.31 | 5.17 | 16.55 |

## H.1 $S$-TABLES: DYNAMIC PROGRAMMING

Consider a MDP where the number of states $|\mathcal{Y}| = 10$. Suppose each state $y$ is associated with a fixed vector $o := [o_1, o_2] \in \mathbb{R}^2$ sampled from a uniform distribution over $[0, 10]^2$. The observations are listed in Table 3. Given a deterministic policy $\pi$ that maps the current state to a next state, and the fixed initial state $y_0$, consider the value

$$\int_{0 < \tau_2 < T} \int_{0 < \tau_1 < \tau_2} do_1 do_2 \tag{H.1}$$

along the path made by linearly interpolating the sequence of $o$. Through dynamic programming of $S$-function, we obtain a $S$-*table* of the signature element corresponding to the value (H.1). The table is shown in Table 4. The $S$-value of each state for time $0$ computed by dynamic programming is indeed the same as what is computed directly by rollout.

Next, using the same setup as above, we suppose that the reward at a state is the sum of the two observations $o_1$ and $o_2$ (see Table 3); for the same deterministic policy, we construct the value table and compare it to the $S$-table created by the interpolation and the transformation of paths presented in Section E. Both are indeed the same, and are shown in Table 5.

As an example, the transformed path from an initial state "2" is given in the left side of Figure 7.

**Chen optimality:** To see Chen optimality, suppose there are only three policies (i.e., $|\Pi| = 3$) for simplicity. All of the three policies are the same except for the transition at state "8"; which are to "1", "2" and "3", respectively. Starting from the initial state "2", the three paths up to time step 5 are given by $2 \to 10 \to 8 \to 1 \to 5 \to 7$, $2 \to 10 \to 8 \to 2 \to 10 \to 8$, and $2 \to 10 \to 8 \to 3 \to 6 \to 9$. Then, we see that Chen optimality equation (E.2) holds for $t = 2$ and state "8", and the optimal cost is equal to $0.85$.

Table 4: $S$-table of the value (H.1) rounded off to two decimal places for the example MDP path from each state and over time horizon 5.

| | 1 | 2 | 3 | 4 | 5 | 6 | 7 | 8 | 9 | 10 |
|---|---|---|---|---|---|---|---|---|---|---|
| 0 | -7.17 | -6.10 | -20.00 | -2.27 | 37.55 | -5.14 | -4.34 | -0.38 | -14.38 | -5.23 |
| 1 | -15.79 | -1.80 | -19.32 | -9.84 | 11.84 | -6.23 | 16.11 | -2.67 | -13.05 | 3.04 |
| 2 | 2.03 | -3.43 | -14.11 | -14.15 | 20.73 | 1.33 | 3.79 | -3.43 | -12.84 | -3.43 |
| 3 | -0.54 | -2.67 | 12.21 | -6.65 | -2.02 | -4.18 | 6.40 | 3.04 | -7.66 | -1.80 |
| 4 | 9.73 | 1.63 | -4.82 | -13.32 | 2.24 | -3.54 | -1.81 | -0.76 | -9.48 | 6.47 |
| 5 | 0.00 | 0.00 | 0.00 | 0.00 | 0.00 | 0.00 | 0.00 | 0.00 | 0.00 | 0.00 |

Table 5: $S$-table for the transformed path of rewards from each state and over time horizon 5. This is the same as the value table computed using Bellman update.

| | 1 | 2 | 3 | 4 | 5 | 6 | 7 | 8 | 9 | 10 |
|---|---|---|---|---|---|---|---|---|---|---|
| 0 | 62.20 | 63.97 | 54.20 | 50.76 | 49.03 | 56.39 | 43.20 | 66.89 | 61.75 | 59.65 |
| 1 | 53.61 | 54.65 | 40.23 | 32.42 | 43.86 | 45.61 | 35.07 | 50.33 | 51.22 | 47.41 |
| 2 | 43.09 | 38.10 | 21.89 | 24.28 | 35.27 | 31.65 | 29.90 | 38.10 | 40.44 | 38.10 |
| 3 | 32.30 | 25.87 | 13.76 | 19.11 | 24.75 | 13.30 | 21.31 | 28.79 | 26.50 | 21.55 |
| 4 | 18.34 | 16.55 | 8.60 | 10.53 | 13.96 | 5.17 | 10.78 | 12.23 | 8.14 | 9.31 |
| 5 | 0.00 | 0.00 | 0.00 | 0.00 | 0.00 | 0.00 | 0.00 | 0.00 | 0.00 | 0.00 |

## H.2 ERROR EXPLOSIONS

We also elaborate on error explosion issues. To see an approximation error on one-step dynamics could lead to error explosions along time steps in terms of signature values, suppose that the ground truth dynamics is given by $x_{t+1} = f(x_t) := [x_{t,1}^{1.1}, x_{t,2}^{1.1}]$ within the state space $\mathbb{R}^2$. Suppose also that the learned dynamics is $\hat{f}(x) = f(x) + [\epsilon, \epsilon]$ where $\epsilon = 0.1$. Let $S_{10}(\sigma)$ be the signatures (up to depth 10) of the path from the initial state $[2.0, 1.2] \in \mathbb{R}^2$ up to (discrete) time steps 10; and $S_{10}(\hat{\sigma})$ be the signatures of the path generated by $\hat{f}$. On the other hand, suppose the approximated signatures are given by $\hat{S} = S_{10}(\sigma) + (\epsilon, \ldots, \epsilon)$. Then, treating $T^{10}(\mathbb{R}^2)$ as a vector in $\mathbb{R}^{2+2^2+\ldots+2^{10}}$, we compare the Euclidean norm errors of $S_{10}(\hat{\sigma})$ and $\hat{S}$ against $S_{10}(\sigma)$. The results are $4.52$ and $147.96$; which imply that the one-step dynamics based approximation could lead to much larger errors on signatures of the expected future path. Note, we assumed that each scalar output suffers from an error $\epsilon$, which may not be the best comparison.

Although it is not required in Chen formulation, we reconstruct the path from the erroneous signatures and compare it against the path following $\hat{f}$. We use three nodes (including the fixed initial node) to reconstruct the path by minimizing the Euclidean norm of the difference between the erroneous signatures and the signatures of the path generated by linearly interpolating the candidate three nodes (with signature depth 10). We use Adam optimizer (Kingma & Ba, 2014) with step size 0.3 and execute 100 iterations. The paths are plotted again in this appendix for reference in the right side of Figure 7; the reconstructed path is still close to the ground-truth one.

## H.3 GENERATING SIMILAR PATHS

Here, we present an application of signature cost to similar path generations (which we did not mention in the main text). To this end, we define the operator $\diamond$ by

$$\alpha \diamond A := (a_0, \alpha a_1, \alpha^2 a_2, \ldots)$$

for $A \in T((\mathcal{X}))$. Now, given a reference path $\sigma_0$ in the space $\mathbb{R}^4$ which represents a path over $x, y$ positions and *difference* to the next positions, mimicking velocity, we consider generating a path $\sigma^*$ with $\alpha > 0$ which minimizes the cost

$$\|S_4(\sigma^*) - \alpha \diamond S_4(\sigma_0)\|_{\mathbb{R}^{d+d^2+\ldots+d^4}}^2.$$

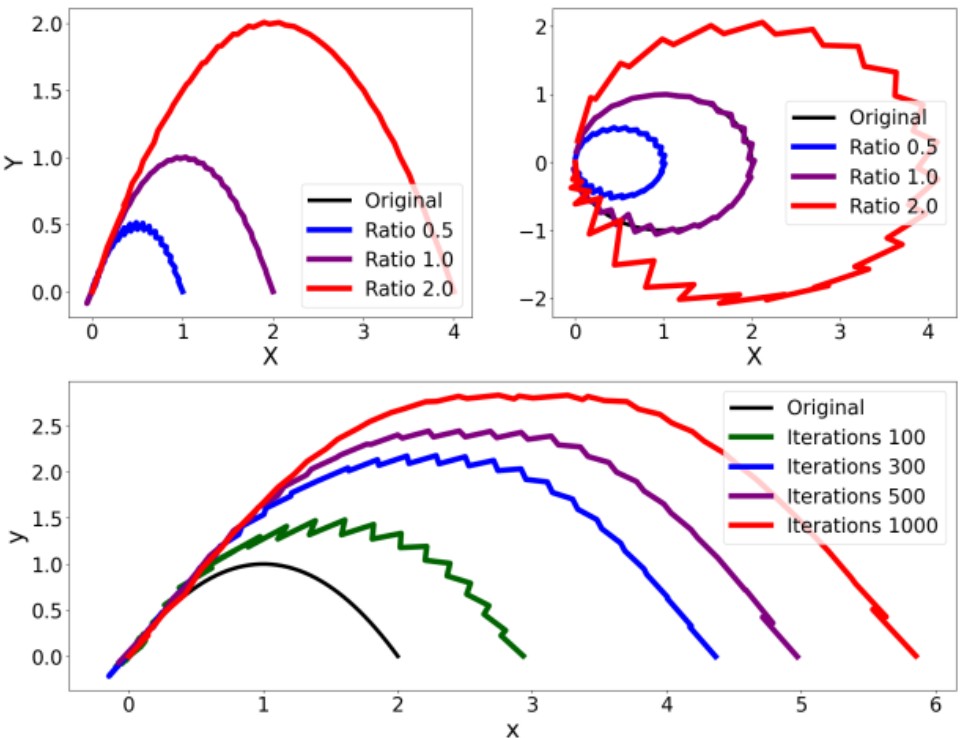

Figure 8: Top: setting different scale $\alpha$ and generating respective optimal path which is expected to be similar to the reference path. Left is for a parabola curve and Right is for a circle. Down: as gradient-based optimization iteration number increases, the generated path scales up while being similar to the orignal path.

We use Adam optimizer with step size $0.1$ and update iterations $500$, and the generated path is an interpolation of 50 nodes. Figure 8 Top shows the optimized paths for different scales of $\alpha$. Using the difference (or velocity) term is essential to recover an accurate path with only the depth $3$ or $4$.

Besides, when the cost is a weighted sum of deviation from the scaled signature and the scale factor itself, we see that as optimization progresses the generated path scales up while being similar to the original reference path. In particular, we use the cost

$$\|S_m(\sigma^*) - \alpha \diamond S_m(\sigma_0)\|^2_{\mathbb{R}^{d+d^2+\dots+d^m}} - \alpha,$$

by treating $\alpha$ as a decision variable as well. It uses the same parameters as above except that we use depth $3$ here. The result is plotted in Figure 8 Down, where the generated paths after gradient steps $100, 300, 500, 1000$ are shown.

## H.4 Additional simple analysis of signatures

Given two different paths $\sigma_1 : [0, T] \to \mathcal{X}$ and $\sigma_2 : [0, T] \to \mathcal{X}$, we plot the squared Euclidean distance between the signatures of those two paths up to each time step, i.e., $\|S(\sigma_1|_{[0,t]}) - S(\sigma_2|_{[0,t]})\|^2$ for $t \in [0, T]$. We test linear paths ($T = 2.0$ and two paths $x = 0.5t$, $x = -0.3t$) and sinusoid paths ($T = 2.0$ and two paths $x = \sin(t\pi)$, $x = \sin(2t\pi)$), and we consider two different base kernels (linear and RBF with bandwidth $0.5$), and two cases, namely the 1D case where $\mathcal{X} = \mathbb{R}$ and the 2D case where $\mathcal{X} = \mathbb{R}^2$ which is augmented with time. We use dyadic order 3 for computing the PDE kernel. The plots are given in Figure 9. From the figure, we see that 1D cases only depend on the start and end points, which confirms the theory; and for 2D cases, the first depth signature terms are still dominant.

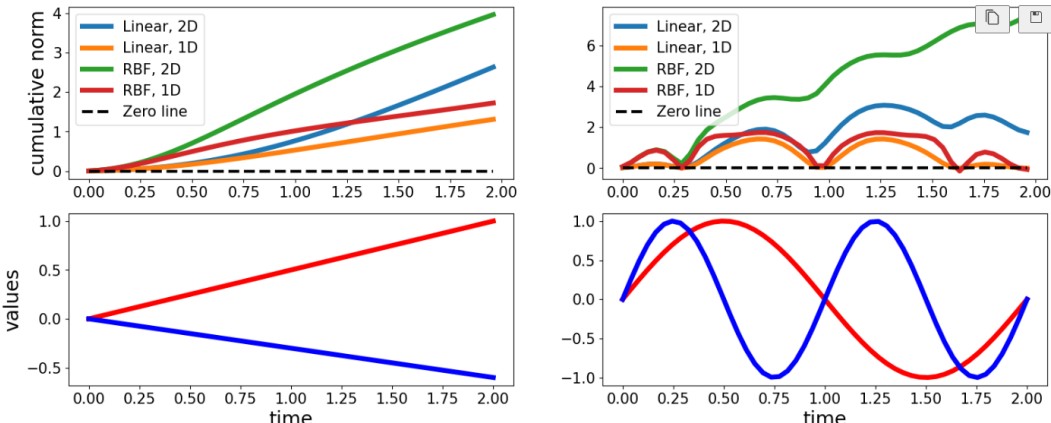

Figure 9: Left: linear paths comparison. Right: sinusoid paths comparison. Top: squared norm of the cumulative difference; for linear base kernel, RBF kernel, and for the 1D case and 2D case. Down: illustrations of two different paths.

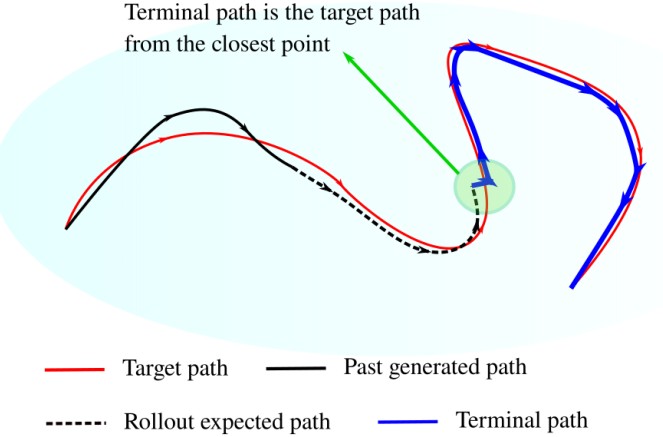

Figure 10: Illustration of the terminal $S$-function used in this work.

Also, using RBF kernels (with narrow bandwidths), the deviation of a pair of paths becomes clarified even if they are close in the original Euclidean space.

# I DETAILS ON THE CHOICE OF TERMINAL $S$-FUNCTIONS

In this section, we present the details of the choice of terminal $S$-functions. Other than the one used in the main text (illustrated in Figure 10), another example of $\mathcal{TS}_m$ is given by

$$\mathcal{TS}_m(x, s, \sigma) \in \underset{u \in T^m(\mathcal{X})}{\arg\min} \ell\left(s \otimes_m S_m(\sigma) \otimes_m u\right) + \ell_{\text{reg}}(u). \tag{I.1}$$

If this computation is hard, one may choose $\mathcal{TS}_m(x, s, \sigma) = \mathbf{1}$; or for path tracking problem, one may choose the signature of a straghtline between the endpoint of $\sigma^*$ and the endpoint of $\sigma$. These three examples are shown in Figure 11

**Terminal $S$-function and surrogate costs:** For an application to MPC problems, we analyze the surrogate cost $\ell$, regularizer $\ell_{\text{reg}}$, and the terminal $S$-function $\mathcal{TS}$ in Algorithm 1. Suppose the problem is to track a given path with signature $s^*$ ($m = \infty$). Fix the cost $\ell$ to $\ell(s) = \|s - s^*\|^2 - w_1\|s\|^2$ and $\ell_{\text{reg}}$ to $\ell_{\text{reg}}(s) = w_2\|s\|^2$, where $w_1, w_2 \in \mathbb{R}_{\geq 0}$ are weights.

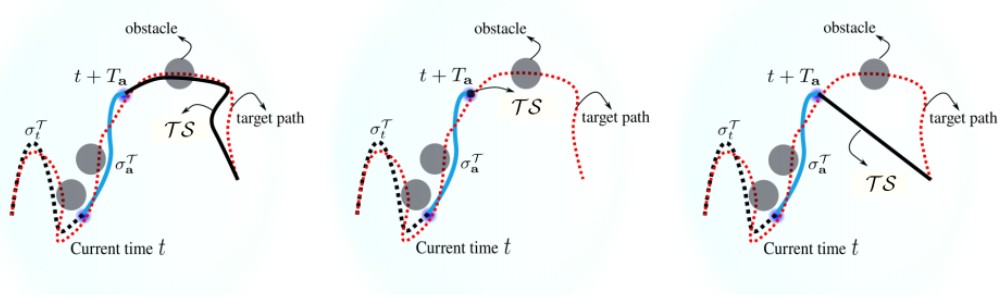

Figure 11: Illustrations of example terminal $S$-functions when given target path to track. $\sigma_t^{\mathcal{T}}$ is the transformed past path whose signature is $s_t$. The left shows (I.1); the future path ignores any dynamic constraints and the optimal virtual path is computed. The middle is for the case that the signature is $\mathbf{1}$. The right is the case where straightline between the endpoint of the target path and the state at time $t + T_{\mathbf{a}}$ is the terminal path.

Here, $\ell_{\mathrm{reg}}$ regularizes so that the terminal path becomes shorter, i.e., the agent prefers progressing more with accuracy sacrifice. The term $w_1\|s\|^2$ for $\ell$ is used to allow some deviations from the reference path.

**Fact I.1** (cf. (Hambly & Lyons, 2010; Boedihardjo & Geng, 2019)). *For the signature $S(\sigma) = (1, s_1, s_2, \ldots)$ of a path $\sigma$ of finite variation on $\mathcal{X}$ with the length $|\sigma| < \infty$, it follows that*

$$\|s_k\|_{\mathcal{X}^{\otimes k}} \leq \frac{|\sigma|^k}{k!}.$$

*For sufficiently* well-behaved *path (see (Hambly & Lyons, 2010; Boedihardjo & Geng, 2019) for example), the limit exists:*

$$\lim_{k \to \infty} \||\sigma|^{-k} k! s_k\|_{\mathcal{X}^{\otimes k}}^2 \leq 1.$$

*If the norm is the projective norm, the limit is $1$.*

From this, we obtain

$$\|S(\sigma)\|^2 = \sum_{k=0}^{\infty} \|s_k\|_{\mathcal{X}^{\otimes k}}^2 \leq \sum_{k=0}^{\infty} \left(\frac{|\sigma|^k}{k!}\right)^2 \leq \left(\sum_{k=0}^{\infty} \frac{|\sigma|^k}{k!}\right)^2 = e^{2|\sigma|},$$

and for a zero length path, i.e., a point, we obtain $1 = \|S(\sigma)\|^2 = e^{2|\sigma|}$. Therefore, while one could use $(k!\|s_k\|_{\mathcal{X}^{\otimes k}})^{1/k}$ for large $k$ as a proxy of $|\sigma|$, we simply use $\|S(\sigma)\|^2$.

We compare the following three different setups with the same surrogate cost and regularizer ($w_1 = 0$, $w_2 = 1$): (1) terminal path is the straightline between the endpoints of the rollout and the reference path, (2) terminal path is computed by nested optimization (see (I.1)), and (3) terminal path is given by the subpath of the reference path from the end time of the rollout. The comparisons are plotted in Figure 12. In particular, for the reference path (linear $x = t$ or sinusoid $x = \sin(t\pi)$) over time interval $[0, 3]$, we use 20 out of 50 nodes to generate subpaths upto the fixed time 1.2. We use Adam optimizer with step size 0.1; and 300 update iterations for all but the type (2) above, which uses 30 iterations both for outer and inner optimizations.

From the figure, our example costs $\ell$ and $\ell_{\mathrm{reg}}$ properly balance accuracy and length of the rollout subpath.

## J EXPERIMENTAL SETUPS

Here, we describe the detailed setups of each experiment and show some extra results.

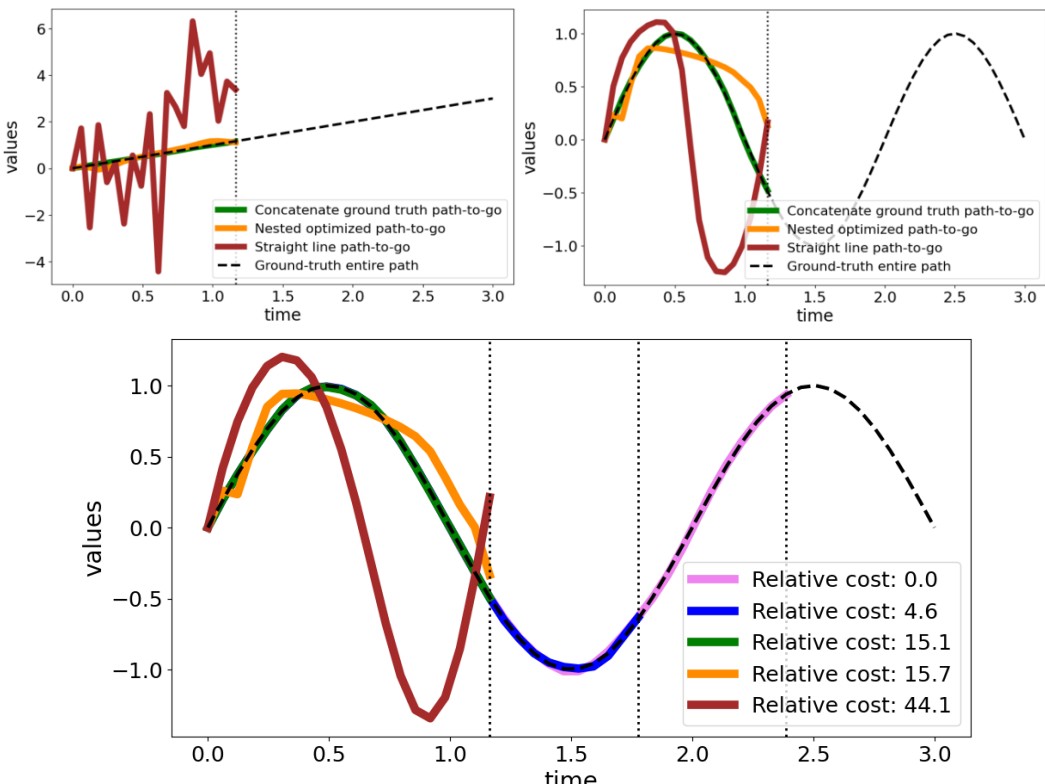

Figure 12: Top: green line is generated by the same approach as the one used in this work; orange one does inner optimization to obtain the optimal terminal $S$-function and the red line uses the straightline as the terminal path. Down: comparison of the costs for the three subpaths and two other longer paths using the same choice of the terminal path as that in this work. As expected, longer subpath has lower score thanks to the regularizer cost.

## J.1 SIMPLE POINTMASS MPC

Define $\mathcal{X} := [0, 100] \times [0, 100] \times [0, 5] \times [0, 5]$. The dynamics is approximated by the Euler approximation:

$$p_{t+1} = P_{\mathcal{X}} \left[ p_t + v_t \Delta t \right],$$

where $p$ is the 2D position and $v$ is the 2D velocity and $P_{\mathcal{X}} : \mathbb{R}^4 \to \mathcal{X}$ is the orthogonal projection.

A feasible reference path for the obstacle avoidance goal reaching task is generated by RRT[*] with local CEM planner (*wiring* of nodes is done through CEM planning with some margin). The parameters used for RRT[*] and CEM planner are shown in Table 6. The generated reference path is shown in Figure 13 Left.

The reference path is then splined by using natural cubic spline (illustrated in Figure 14; using package (`https://github.com/patrick-kidger/torchcubicspline`); using only the path over positions (2D path), we run signature MPC. The time duration of each action is also optimized at the same time. For comparsion we also run a MPC with zero terminal $S$-function case. Note we are using RBF kernel for signature kernels, which makes this terminal $S$-function choice less unfavorable.

Figure 13 Middle shows the zero terminal $S$-function case; which tracks well but with slight deviation. Right shows that of the best choice of terminal $S$-function.

The parameters for the signature MPC is given in Table 7. Here, scaling of the states indicates that we multiply the path by this value and then compute the cost over the scaled path. Note we used

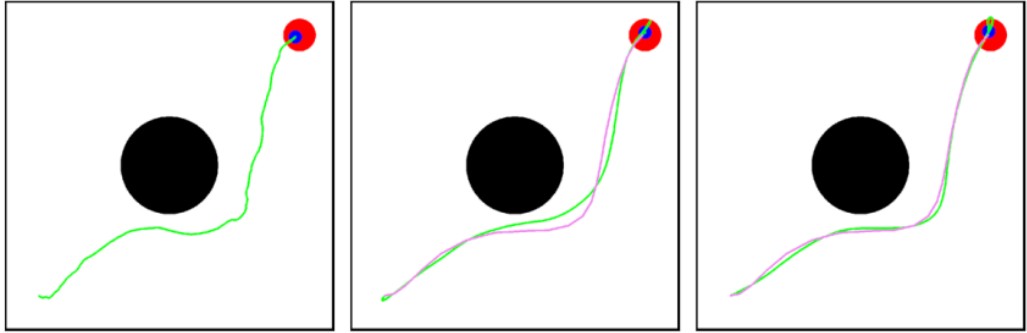

Figure 13: Left: suboptimal feasible path generated by RRT* with local CEM planner. Middle: signature MPC with zero terminal $S$-function. Purple is the splined reference path and the green is the executed one. Right: signature MPC with the best choice of terminal $S$-function.

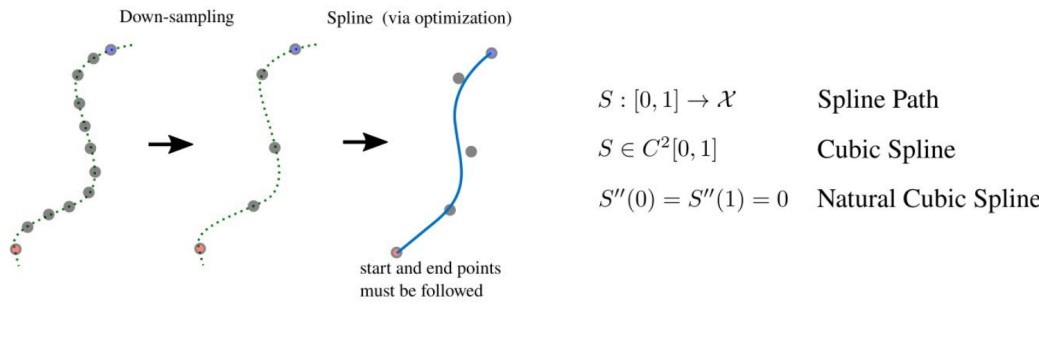

Figure 14: Illustrations of natural cubic spline. We down-sampled the reference path with skip number 10. Then, with weight $w = 0.5$, step size $0.01$, using Adam optimizer, we optimized the path to obtain a spline with iteration number 150.

torchdiffeq package (Chen et al., 2018; 2021) of PyTorch (Paszke et al., 2017) to compute rollout, and the evaluated points are of switching points of actions, and it replans when the current action repetition ends.

## J.2 INTEGRAL CONTROL EXAMPLES

Our continuous time system of two-mass, spring, damper system is given by

$$\dot{v_1} = -\frac{(k_1 + k_2)p_1}{m_1} - \frac{(b_1 + b_2)v_1}{m_1} + \frac{k_2 p_2}{m_1} + \frac{b_2 v_2}{m_1} + \frac{a_1}{m_1} + w_1,$$

$$\dot{v_2} = \frac{k_2 p_1}{m_2} + \frac{b_2 v_1}{m_2} - \frac{k_2 p_2}{m_2} - \frac{b_2 v_2}{m_2} + \frac{a_2}{m_1} + w_2,$$

where $u_1, u_2 \in [-1, 1]$ are control inputs, and $ws$ are disturbances. The actual parameters are listed in Table 8. We augment the state with time that obviously follows $\dot{t} = 1$. We obtain an approximation of the time derivative of signatures by

$$\frac{\partial S_2(\sigma_t)}{\partial t} \approx \frac{S_2(\sigma_t * \sigma_{t,t+\Delta t}) - S_2(\sigma_t)}{\Delta t},$$

Table 6: RRT[*] and CEM parameters.

| max distance to the sample | 10.0 | goal state sample rate | 0.2 |
|---|---|---|---|
| safety margin to obstacle | 0.0 | $\gamma$ to determine neighbors | 1.0 |
| CEM distance cost | quadratic | CEM obstacle penalty | 1000 |
| CEM elite number | 3 | CEM sample number | 8 |
| CEM iteration number | 3 | numpy random seed | 1234 |

Table 7: Parameters for the signature MPCs of point-mass (shared values for zero terminal $S$-function and the best choice ones).

| static kernel | RBF/scale 0.5 | dyadic order of PDE kernel | 2 |
|---|---|---|---|
| scaling of the states | 0.05 | update number of PyTorch | 20 |
| step size for update | 0.2 | number of actions $N$ | 3 |
| weight $w_1$ | 0 | regularizer weight $w_2$ | 8.0 |
| maximum magnitude of control | 1.0 | | |

where $\sigma_t$ is the path from time 0 to $t$, and $\sigma_{t,t+\Delta t}$ is the linear path between the current state and the next state (after $\Delta t$), and $\Delta t = 0.1$ is the discrete time interval.

By further augmenting the state with signature $S_2(\sigma_t)$, we compute the linearized dynamics around the point $p = v = \mathbf{0}_2$ and the unit signature.

P control (the state includes velocities, so it might be viewed as PD control) is obtained by computing the optimal gain for the system over $p$ and $v$ by using the cost $p^\top p + v^\top v + 0.01 u^\top u$. For PI control, we use the linearized system over $p$, $v$ and $s_{2,5}, s_{2,15}$ (corresponding to integrated errors for $p_1$ and $p_2$), and the cost is $p^\top p + v^\top v + s_{2,5}^2 + s_{2,15}^2 + 0.01 u^\top u$. We added $-0.0001I$ to the linearized system to ensure that the python control package (https://github.com/python-control/python-control) returns a stabilizing solution.

The unknown constant disturbance is assumed zero for planning, but is $0.03$ (on both acceleration terms) for executions; the plots are given in Figure 15. It reflects the well-known behaviors of P control and PI control.

We also list the parameters used for signature MPCs described in Section 6; the execution horizon is $15.0$ sec, and the reference is the signature of the linear path over the zero state along time interval from 0 to $25.0$ (we extended the reference from $15.0$ to $25.0$ to increase stability). The control inputs are assumed to be fixed over planning horizon, and are actually executed over a planning interval. The number of evaluation points when planning is given so that the rollout path is approximated by the piecewise linear interpolation of those points (e.g., for planning horizon of $1.0$ sec with 5 eval points, a candidate of rollout path is evaluated evenly with $0.2$ sec interval).

The signature cost is the squared Euclidean distance between the reference path signature and the generated path signature upto depth 1 or 2; the terminal path is just a straighline along the time axis, staying at the current state.

The parameters used for signature MPCs are listed in Table 9; note the truncation depths for signatures are 1 and 2, respectively. The plots are given in Figure 15.

## J.3 PATH TRACKING WITH ANT

We use DiffRL package (Xu et al., 2021) Ant model.

**Reference generation:** We generate reference path over 2D plane with the points

$$[0.0, 0.0], [2.6, -3.9], [5.85, -1.95], [6.5, 0.0], [5.85, 1.95],$$
$$[2.6, 3.9], [0.0, 3.51], [-3.25, 0.0], [-6.5, -3.9], [-6.5, 4.55]$$

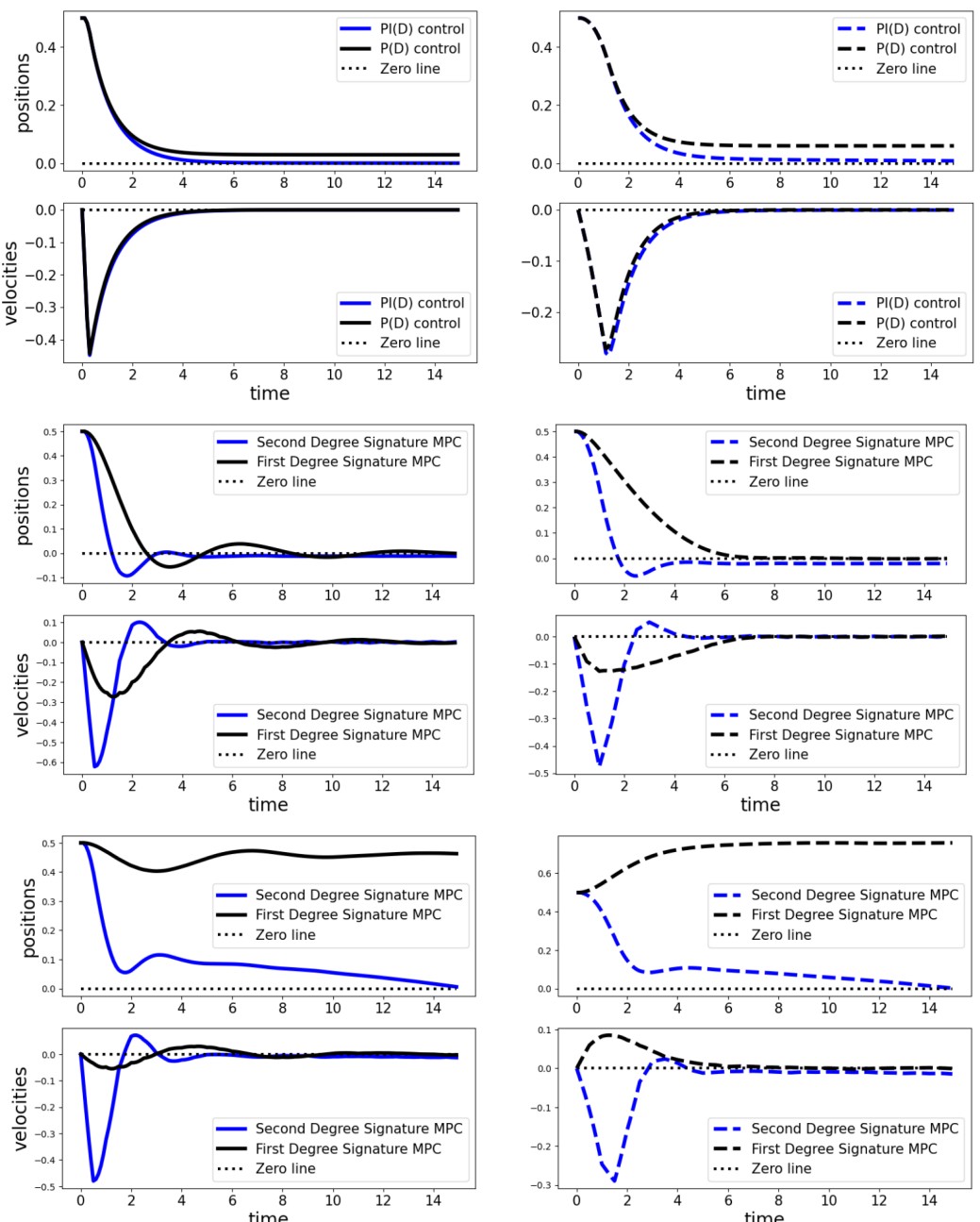

Figure 15: Two-mass spring, damper system. Solid lines are for the position and velocity of the first mass, and dashed lines are for the second mass. Black lines are without signatures, and the blue lines are with the elements of the second degree signatures corresponding to the integrated errors. Top: P control and PI control under no disturbance. Middle: signature MPCs using signatures upto depth 1 and 2 under no disturbance case; both converge to the zero state well. Down: signature MPCs under unknown disturbances; the one upto depth 2 converges to zero state while the one upto depth 1 does not because the depth 2 signature terms correspond to the integrated errors to mimic PI controls.

and we obtain the splined path with 2000 nodes (skip number 1, weight for smoothness 0.5, iteration number 150 with Adam optimizer). Also, to use for the terminal path in MPC planning, we obtain rougher one with 200 nodes.

Table 8: Parameters for two-mass, spring, damper system.

| $k_1$ | 2.0 | $b_1$ | 0.05 | $m_1$ | 1.0 |
|---|---|---|---|---|---|
| $k_2$ | 1.0 | $b_2$ | 0.05 | $m_2$ | 2.0 |

Table 9: Parameters for the signature MPC.

| kernel type | truncated linear | horizon for MPC | 1.0 sec |
|---|---|---|---|
| number of eval points | 5 | update number per step | 50 |
| step size for update | 0.1 | planning interval | 0.5 sec |
| number of actions | 1 | max horizon | 25.0 sec |
| maximum magnitude of control | 1.0 | | |

We run signature MPC (parameters are listed in Table 10) and the simulation steps to reach the endpoint of the reference is found to be 880. Then, we run the baseline MPC; for the baseline, we generate 880 nodes for the spline (instead of 2000). Note these waypoints are equally sampled from $t = 0$ to $t = 1$ of the obtained natural cubic spline. We also test slower version of baseline MPC with 1500 and 2500 simulation steps (i.e., number of waypoints).

**Cost and reward:** The baseline MPC uses time-varying waypoints by augmenting the state with time index. The time-varying immediate cost $c_{\text{baseline}}$ to use is inspired by Peng et al. (2018):

$$c_{\text{baseline}}(x, t) = -\sum_{i=1}^{d} \exp\{-10(x_{(i)} - x_{(i)}^*(t))^2\}$$

for time step $t$, where $x^*(t)$ is the (scaled) waypoint at time $t$ and $x_{(i)}$ is the $i$th dimension of (the scaled state) $x \in \mathbb{R}^d$. In addition to this cost, we add height reward for the baseline MPC:

$$r_{\text{height}}(z) = -100\text{LeakyReLU}_{0.001}(0.37 - z),$$

where $\text{LeakyReLU}_{0.001}$ is the Leaky ReLU function with negative slope 0.001 and $z$ is the height of Ant. For signature MPC, in addition to the signature cost described in the main text, we also add Bellman reward $10r_{\text{height}}$ (the scale 10 is multiplied to balance between signature cost and the height reward).

**Experimental settings and evaluations:** The parameters are listed in Table 10. The parameters used for SAC RL are listed in Table 11.

Since the model we use is differentiable, we use Adam optimizer to optimize rollout path by computing gradients through path. Surprisingly, with only 3 gradient steps per simulation step, it is working well; conceptually, this is similar to MPPI (Williams et al., 2017) approach where the distribution is updated once per simulation step and the computed actions are shifted and kept for the next planning. Also, we set the maximum points of the past path to obtain signatures to 50 (we skip some points when the past path contains more than 50 points).

To evaluate the accuracy, we generate 2000 nodes from the spline of the reference, and we compute the Euclidean distance from each node of the reference to the closest simulated point of the generated trajectory. The relative cumulative deviations are plotted in Figure 16. For the same reaching time, signature MPC is significantly more accurate. When the speed is slowed, baseline MPC becomes a bit more accurate. Note our signature MPC can also tune the tradeoff between accuracy and progress without knowing feasible waypoints.

Also, the performance curve of SAC RL is plotted in Figure 17 Left against the cumulative reward achieved by the signature MPC counterpart. We see that RL shows poor performance (refer to the discussions on difficulty of RL for path tracking problems in (Peng et al., 2018)).

Table 10: Parameters for the signature MPC for Ant. Baseline MPC shares most of the common values except for scaling, which is 1.0 for baseline MPC.

| static kernel | RBF/scale 0.5 | dyadic order of PDE kernel | 1 |
|---|---|---|---|
| scaling of the states | 0.2 | update number of PyTorch | 3 |
| step size for update | 0.1 | number of actions $N$ | 64 |
| weight $w_1$ | 0 | regularizer weight $w_2$ | 3.0 |
| maximum magnitude of control | 1.0 | | |

Table 11: Parameters for the SAC RL for Ant path following.

| number of steps per episode | 128 | initial alpha for entropy | 1 |
|---|---|---|---|
| step size for alpha | 0.005 | step size for actor | 0.0005 |
| step size for $Q$-function | 0.0005 | update coefficient to target $Q$-function | 0.005 |
| replay buffer size | $10^6$ | number of actors | 64 |
| NN units for all networks | $[256, 128, 64]$ | batch size | 4096 |
| activation function for NN | tanh | episode length | 1000 |

### J.4 PATH TRACKING WITH FRANKA ARM END-EFFECTOR

We use DiffRL package again and a new Franka arm model is created from URDF model (Sutanto et al., 2020).

**Model:** The stiffness and damping for each joint are given by

$$\text{stiffness} : 400, 400, 400, 400, 400, 400, 400, 10^6, 10^6,$$
$$\text{damping} : 80, 80, 80, 80, 80, 80, 80, 100, 100,$$

and the initial positions of each joint are

$$1.157, -1.066, -0.155, -2.239, -1.841, 1.003, 0.469, 0.035, 0.035.$$

The action strength is $60.0 \, \text{N} \cdot \text{m}$. Simulation step is $1/60$ sec and the simulation substeps are 64.

**Reference generation:** We similarly generate reference path for the end-effector position with the points

$$[0.0, 0.0, 0.0], [0.1, -0.1, -0.1], [0.2, -0.15, -0.2], [0.18, 0.0, -0.18],$$
$$[0.12, 0.1, -0.12], [0.08, -0.1, 0.0], [0.05, -0.15, 0.1], [0.0, -0.12, 0.2],$$
$$[-0.05, -0.05, 0.25], [-0.1, 0.1, 0.15], [-0.05, 0.05, 0.08]$$

and we obtain the splined path with 1000 nodes (skip number 1, weight for smoothness 0.001, iteration number 150 with Adam optimizer). Also, to use for the terminal path in MPC planning, we obtain rougher one with 100 nodes. Similar to Ant experiments, we use 270 equally assigned waypoints for the baseline MPC.

For the case with unknown disturbance, we use the same waypoints.

**Experimental settings and evaluations:** The parameters for MPCs are listed in Table 13. The parameters used for SAC RL are listed in Table 14.

We again set the maximum points of the past path to obtain signatures to 50.

The unknown disturbances are added to *every* joint.

We list the results of all the disturbance cases in Table 12 as well. Also, the performance curve of SAC RL is plotted in Figure 17 Right against the cumulative reward achieved by the signature MPC counterpart. We see that the cumulative reward itself of SAC RL outperforms the signature MPC

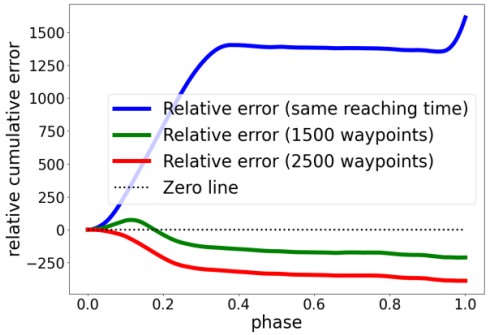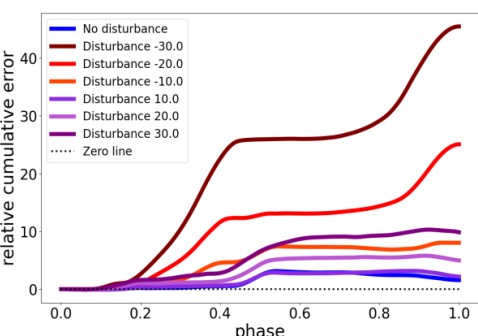

Figure 16: For each node of the reference path, we compute the Euclidean distance from the closest simulated point of the generated trajectory. Phase is from 0 to 1, corresponding to the start and end points of the reference. Left shows the relative *cumulative* deviations along 2000 nodes of the reference for Ant. It is the errors of the baseline MPC compared to the errors of signature MPC; hence positive value shows the advantage of signature MPC. Right shows those for robotic arm experiments with different magnitudes of disturbances added to every joint.

Table 12: All results on path tracking with a robotic manipulator end-effector. Comparing signature control, and baseline MPC and SAC RL with equally assigned waypoints under unknown fixed disturbance.

|  | | Deviation (distance) from reference | |
|---|---|---|---|
|  | Disturbance ($N \cdot m$) | Mean ($10^{-2}$m) | Variance ($10^{-2}$m) |
| signature control | $+30$ | **1.674** | **0.002** |
|  | $+20$ | **1.022** | **0.002** |
|  | $+10$ | **0.615** | **0.001** |
|  | $\pm0$ | **0.458** | **0.001** |
|  | $-10$ | **0.605** | **0.001** |
|  | $-20$ | **0.900** | **0.001** |
|  | $-30$ | **1.255** | **0.002** |
| baseline MPC | $+30$ | 2.648 | 0.015 |
|  | $+20$ | 1.513 | 0.010 |
|  | $+10$ | 0.828 | 0.005 |
|  | $\pm0$ | 0.612 | 0.007 |
|  | $-10$ | 1.407 | 0.013 |
|  | $-20$ | 3.408 | 0.078 |
|  | $-30$ | 5.803 | 0.209 |
| SAC RL | $+30$ | 15.669 | 0.405 |
|  | $+20$ | 10.912 | 0.224 |
|  | $+10$ | 6.252 | 0.102 |
|  | $\pm0$ | 3.853 | 0.052 |
|  | $-10$ | 6.626 | 0.243 |
|  | $-20$ | 12.100 | 0.557 |
|  | $-30$ | 16.019 | 0.743 |

for no disturbance case for the robotics arm experiment; however it does not necessarily show better tracking accuracy along the path.

To evaluate the accuracy, we generate 1000 nodes from the spline of the reference. The relative cumulative deviations are plotted in Figure 16. For all of the disturbance magnitude cases, signature MPC is more accurate. When the disturbance becomes larger, this difference becomes significant, showing robustness of our method.

Visually, the generated trajectories are shown in Figure 18.

Table 13: Parameters for the signature MPC for robotic arm. Baseline MPC shares most of the common values except for scaling, which is 1.0 for baseline MPC.

| static kernel | RBF/scale 0.5 | dyadic order of PDE kernel | 1 |
|---|---|---|---|
| scaling of the states | 10.0 | update number of PyTorch | 3 |
| step size for update | 0.1 | number of actions $N$ | 16 |
| weight $w_1$ | 0.5 | regularizer weight $w_2$ | 0.5 |
| maximum magnitude of control | 1.0 | | |

Table 14: Parameters for the SAC RL for robotic arm path following.

| number of steps per episode | 128 | initial alpha for entropy | 1 |
|---|---|---|---|
| step size for alpha | 0.005 | step size for actor | 0.0005 |
| step size for $Q$-function | 0.0005 | update coefficient to target $Q$-function | 0.005 |
| replay buffer size | $10^6$ | number of actors | 64 |
| NN units for all networks | $[256, 128, 64]$ | batch size | 2048 |
| activation function for NN | tanh | episode length | 300 |

## K POTENTIAL APPLICATIONS TO REINFORCEMENT LEARNING

Although we have not yet developed promising RL algorithm; the basic signature RL algorithm is summarized in Algorithm 2. In general, the algorithm stores states, actions, and past signatures up to depth $m$ observed in rollout trajectories, and update $S$-function using Chen formulation followed by a policy update.

Especially, in this work, we use the modified version of soft actor-critic (SAC) (Haarnoja et al., 2018) to adapt to signature RL.

**Task:** The task is based on the similar path generation using signature cost. In particular, we test Hopper for long jump. We use DiffRL Hopper model and use rl_games (Makoviichuk & Makoviychuk, 2021) for implementations. The orignal (unit scale) reference path mimics the curve of jumps over $x$-axis/height position and velocity with the points:

$$[0.0, 0.0, 0.0, 0.0], [0.0, 0.0, 3.0, 3.9], [0.3, 0.39, 3.0, 2.63], [0.6, 0.65, 3.0, 1.35],$$
$$[0.9, 0.79, 3.0, 0.08], [1.2, 0.80, 3.0, -1.20], [1.5, 0.68, 3.0, -2.47], [1.8, 0.43, 3.0, -3.74],$$
$$[2.1, 0.05, 3.0, -5.02], [2.13, 0.0, 3.0, -5.16], [2.13, 0.0, 3.0, 0.0].$$

**Cost:** The cost for the RL task is given by

$$\ell(s_m) = -3000 \left[ 2 \exp\left(-0.1 \| s_m - \alpha \diamond s_m^* \|_1 \right) + \alpha \right],$$

where $s_m^*$ is the (truncated) signature of the unit scale reference path. We use this cost for signature RL and its negative as the reward for SAC based on Signature MDP (where the state is augmented with signatures, and the reward, or negative cost, is added at the terminal time; hence uses traditional Bellman updates). Note that the cost/reward to optimize for signature RL and for SAC are not exactly the same (because of the order of expectation); but for simplicity, we will evaluate the performance based on the reward of Signature MDP. See Section G for details.

**Experimental settings:** We use four neural networks for the $S$-function, signature cost function (using $S$-function), $Q$-function (recall Chen formulation subsumes Bellman reward), and the policy function. The network sizes for them are $[512, 512, 256]$, $[256, 128, 64]$, $[256, 128, 64]$, and $[256, 128, 64]$, respectively; SAC based on Signature MDP also shares the sizes of the common networks. Parameters used for both RL algorithms are listed in Table 15.

**Results:** Signature RL has shown very fast increase of reward at the initial phase but then struggles to improve while SAC based on Signature MDP shows slow but steady growth. We run 5 seeds and

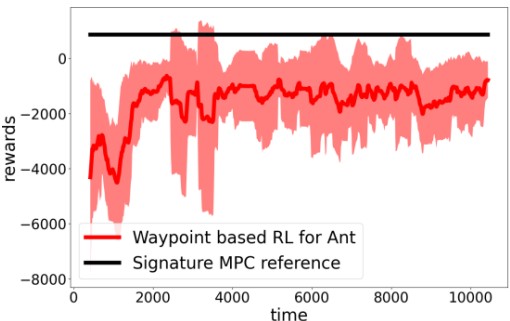 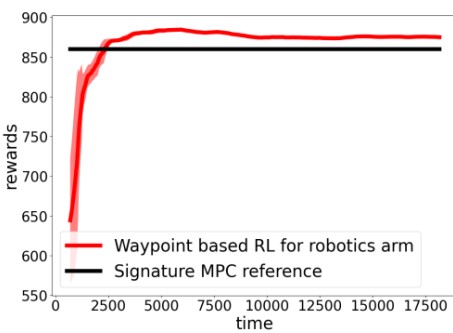

Figure 17: Soft-actor-critic reinforcement learning baseline for an ant (left) and robotic arm (right) path following experiment. The red curve shows the average cumulative rewards with standard deviation shade over five different seed runs, and the black line is for reference of the cumulative rewards achieved by signature control. The reward is the same (negative cost) as that used in the baseline MPC, and the state is augmented with the time step (maximum time step is 1000 and 300 given that the goal-reaching time of signature control is 880 and 270 steps, respectively). For the ant case, RL did not achieve comparable performance in terms of rewards. For the robotics arm case under no disturbance, the RL outperforms signature control slightly.

---

**Algorithm 2** Signature RL

---

  **Input**: initial policy $\pi$; initial signature $s_0 = \mathbf{1}$; depth $m$; initial approximated $S$-function $\hat{\mathcal{S}}_m^\pi$; initial distribution $P_0$ over state space $\mathcal{X}$; time horizon $T$; surrogate cost $\ell$

  **Output**: policy $\pi^*$ and its $S$-function $\mathcal{S}_m^{\pi^*}$

1: **while** not convergent **do**
2:     Sample an initial state $x_0 \sim P_0$
3:     Run current policy $\pi$ to collect trajectory data $\tau := (s_0, x_0, a_0, s_{t_{a_0}}, x_{t_{a_0}}, a_{t_{a_0}}, \ldots, s_T, x_T)$
4:     Compute $m$th-depth signatures $S(\sigma(x_t, x_{t+t_a}))$s for each transformed path connecting a pair of adjacent states and action $a$ in $\tau$
5:     Update $S$-function so that

$$\hat{\mathcal{S}}_m^\pi(a, P_{\mathcal{Y}}(x_t)) \approx \mathbb{E}\left[ S_m(\sigma(x_t, x_{t+t_a})) \otimes_m \hat{\mathcal{S}}_m^\pi(a', P_{\mathcal{Y}}(x_{t+t_a})) | \omega \in \Omega_a \right],$$

    where the expectation is approximated by an arithmetic mean and $a'$ is sampled from $\pi$ at $x_{t+t_a}$
6:     Update policy by, for example, minimizing

$$\ell\left( s \otimes_m \mathbb{E}\left[ \hat{\mathcal{S}}_m^\pi(b(\omega), P_{\mathcal{Y}}(x)) \right] \right)$$

    for all pairs of $(s, x)$s in $\tau$ or in the buffer
7: **end while**
8: Output $\pi$ and $\hat{\mathcal{S}}_m^\pi$.

---

the plots of averaged reward/step growth are shown in Figure 19 with standard deviation shadows. A possible reason for this is because of the lack of guarantees of convergence to the optima; and developing signature RL algorithms that have convergence guarantees and practical scaling is a very important future work.

The generated long jump motion of the best performance of SAC (based on Signature MDP) is shown in Figure 20.

## L    COMPUTATIONAL SETUPS AND LICENCES

For all of the experiments, we used the computer with

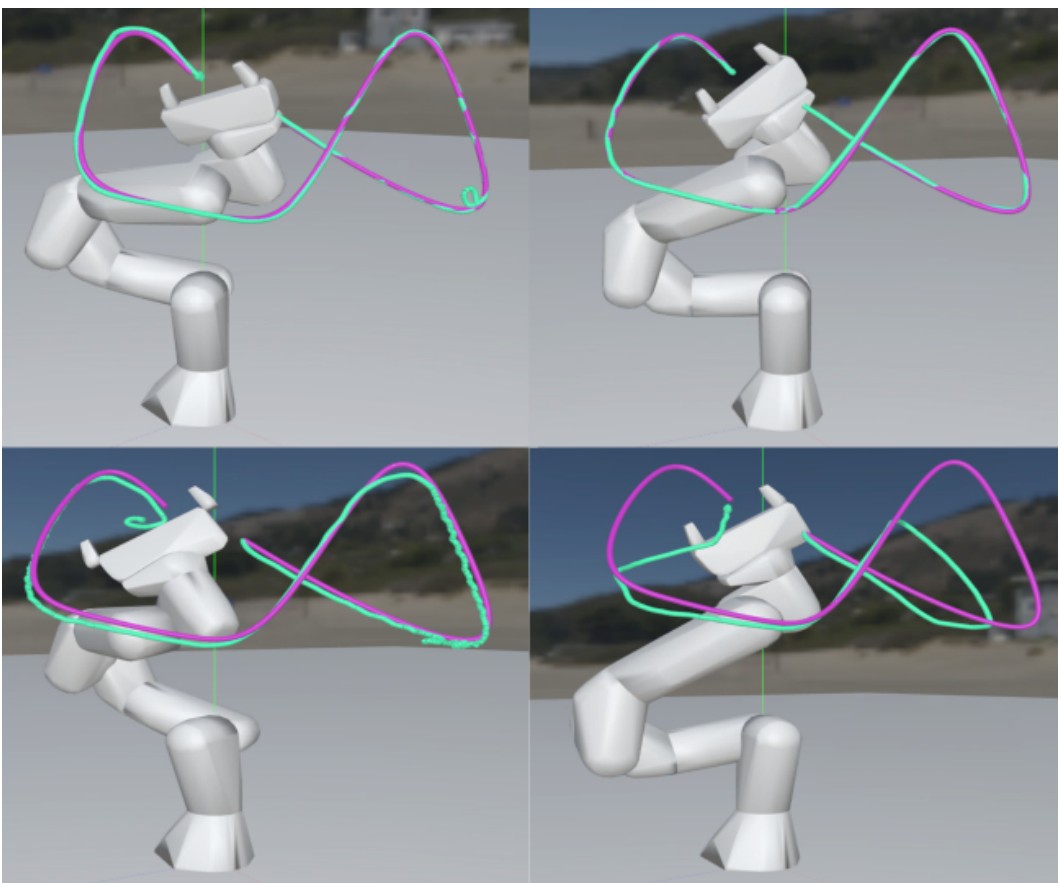

Figure 18: Top: Left shows signature control result for robotic arm manipulator end-effector path tracking and Right shows the baseline MPC. They are tracking the path similarly well. Down: under disturbance $-30.0$. The baseline tracking accuracy is deteriorated largely while signature control is robust against disturbance. Signature MPC tries to track the first curve *stubbornly* by taking time there to retrace better. This is because the signature MPC is insensitive to waypoint designs but rather depends on the "distance" between the target path and the rollout path in the signature space.

Table 15: Parameters for the signature RL and SAC for Hopper.

| | | | |
|---|---|---|---|
| number of steps per episode | 16 | initial alpha for entropy | 1 |
| step size for alpha | 0.005 | step size for actor | 0.0002 |
| step size for $Q$-function | 0.0005 | update coefficient to target $Q$-function | 0.005 |
| step size for $S$-function | 0.0002 | update coefficient to target $S$-function | 0.002 |
| step size for signature cost function | 0.002 | batch size | 2048 |
| replay buffer size | $10^6$ | signature depth | 3 |
| maximum magnitude of control | 1.0 | | |

- Ubuntu 20.04.3 LTS
- Intel(R) Core(TM) i7-6850K CPU  3.60GHz (max core 12)
- RAM 64 GB/ 1 TB SSD
- GTX 1080 Ti (max 4; we used the same GPU for all of the experiments)
- GPU RAM 11 GB
- CUDA 10.1

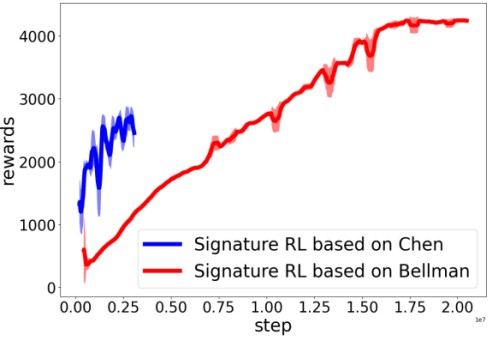 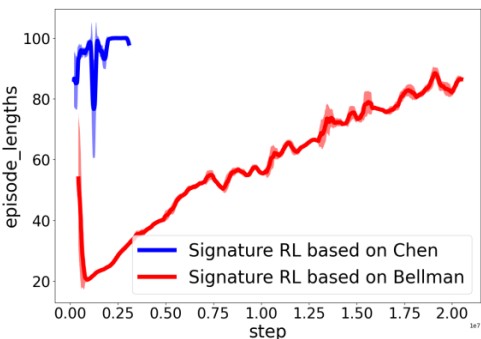

Figure 19: Left shows the growth of trajectory reward along steps and Right shows that of length of the trajectory (the episode terminates when it encounters terminal conditions) along steps. Signature RL has shown very fast increase of reward and length of trajectory at the initial phase but then struggles to improve. The curve is averaged over 5 seed runs and is smoothed.

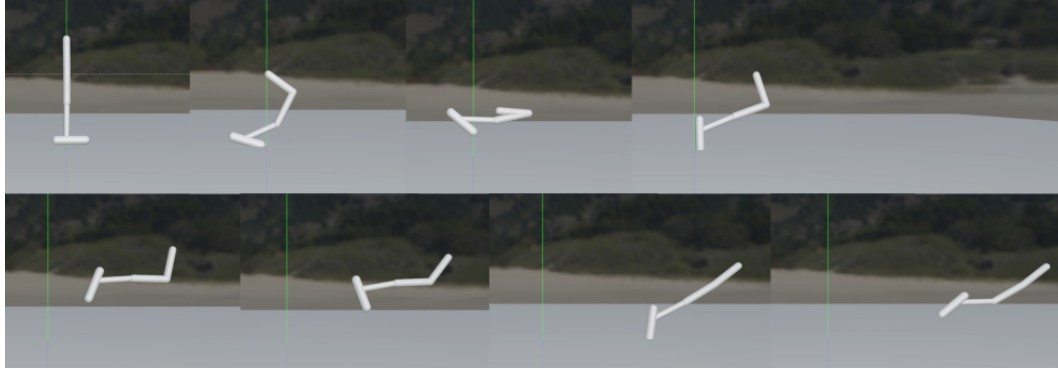

Figure 20: Learned long jump motion of a hopper robot simulation of RL using the signature cost for scaled similar path generation.

The licenses of sigkernel, torchcubicspline, signatory, DiffRL, rl_games, and Franka URDF model, are [Apache License 2.0; Copyright [2021] [Cristopher Salvi]], [Apache License 2.0; Copyright [Patrick Kidger and others]], [Apache License 2.0; Copyright [Patrick Kidger and others]], [NVIDIA Source Code License], [MIT License; Copyright (c) 2019 Denys88], and [MIT License; Copyright (c) Facebook, Inc. and its affiliates], respectively.

The computation time for each (seed) run for any of the numerical experiments was around 30 mins to $1 - 2$ hours (MPC problems). For small analysis experiments, it took less than a few minutes for each one.

