# OpenReview forum: "Signatures Meet Dynamic Programming: Generalizing Bellman Equations for Trajectory Following"
_ICLR.cc/2024/Conference — ICLR 2024 Conference Withdrawn Submission_

### Official Review · Reviewer_ZzLg · 2023-11-01

**Soundness:** 2 fair
**Presentation:** 3 good
**Contribution:** 2 fair
**Rating:** 5
**Confidence:** 3

**Summary:**

The paper establishes connections between value functions used in optimal control and properties of path signatures, which are a powerful representation of paths with useful algebraic properties. Furthermore, they introduce a novel control framework called signature control, which efficiently generalizes the Bellman equation to the space of trajectories. This framework can naturally handle varying/adaptive time steps, propagate higher-level information more efficiently than value function updates, and is robust to dynamical system misspecification over long rollouts. They also devise a model predictive control method for path tracking within the signature control framework. This method generalizes integral control and is suitable for problems with unknown disturbances. The proposed algorithms are tested in simulation with differentiable physics models, including control and robotics tasks.

**Strengths:**

The strengths of the paper are as follows:

1) The paper establishes connections between value functions used in optimal control and properties of path signatures, providing a novel approach to trajectory following.

2) The proposed signature control framework efficiently generalizes the Bellman equation to the space of trajectories, offering advantages such as handling varying/adaptive time steps and propagating higher-level information more efficiently than value function updates . The framework is robust to dynamical system misspecification over long rollouts, making it suitable for real-world applications . The paper presents a model predictive control method for path tracking within the signature control framework, which generalizes integral control and can handle problems with unknown disturbances.

3) The algorithms proposed in the paper are tested in simulation with differentiable physics models, including control and robotics tasks, demonstrating their practical applicability.

4) The paper offers a general approach that lifts the classical Bellman-based dynamic programming to the space of paths, providing more flexibility in devising algorithms without the need for problem-specific modifications

**Weaknesses:**

1) The paper lacks a detailed discussion on the limitations and potential challenges of the signature control framework and the proposed algorithms. It would be beneficial to address any potential drawbacks or scenarios where the framework may not perform optimally.

2) The evaluation of the proposed algorithms is limited to simulation experiments with differentiable physics models. While this provides initial validation, it would be valuable to include real-world experiments or comparisons with existing control methods to demonstrate the effectiveness and practicality of the signature control framework.

3) The paper does not provide a comprehensive analysis of the computational complexity or scalability of the signature control framework. It would be helpful to discuss the computational requirements and potential limitations when applying the framework to more complex and larger-scale problems.

**Questions:**

The paper could benefit from a more thorough discussion on the generalizability of the signature control framework to a wider range of control and robotics tasks. This would provide insights into the applicability of the framework beyond the specific tasks considered in the simulations.

---

### Official Review · Reviewer_1sNc · 2023-11-01

**Soundness:** 2 fair
**Presentation:** 1 poor
**Contribution:** 3 good
**Rating:** 3
**Confidence:** 3

**Summary:**

The authors propose a novel framework, path signature control, that applies path signatures to decision-making for path-tracking problems. Their motivation for this formulation is to address the limitations of the scalar representations of policies learned with dynamic programming-based algorithms in optimal control and reinforcement learning. The authors show the equivalence of their Path signature framework with dynamic programming and propose an MPC-based version of Signature control, which their experiments suggest is more robust to vanilla MPC and SAC under disturbances.

**Strengths:**

The paper offers a new perspective on addressing path-tracking problems with Path Signatures. The work seems promising to generate more robust control algorithms using the author's framework and their connections with dynamic programming help offer new perspectives on solving control problems for path planning.  The authors seems to validate their algorithm across a number of testing environments which is promising to validate their proposed framework and it's instantiation. We appreciate that the authors have included as much information as possible, taking advantage of the appendix. We will note that although we have extensive comments in the weaknesses section, we find the problem the authors quite interesting and the authors general ideas on how to address it quite fascinating.

**Weaknesses:**

This reviewer feels a non-trivial amount of work is needed to improve the quality of the submitted paper before it is accepted. This reviewer found the paper difficult to read, starting with the introduction. The current introduction needs an overhaul to explain better the problem the authors are addressing, why previous solutions do not address this problem, and how their solution addresses these limitations. The introductions' current version needs to clarify that the primary focus is proposing a framework to solve the path-tracking problem. The intro only mentions the path-tracking problem halfway into the second paragraph, treating it as an afterthought of the paper instead of the main problem studied. Information should be provided to the reader describing the importance of the problem and include some description of the problem. The authors should split the content of the second paragraph into at least two or three paragraphs. We strongly advise the authors to reformulate the entire introduction to focus discussion on the problem studied in the paper: addressing the path tracking problem with a novel formulation of path signatures as an alternative framework to dynamic programming, which has the limitations listed in the first paragraph.
Although we have focused our comments on the introductions, other portions of the paper could benefit from improvements in the writing:
- The definitions of path signatures were not easy to follow in section 3.1 and might benefit from a picture of what a signature looks like in practice. It is unclear how one might implement any of these ideas in practice based on the information provided in the paper.
-Furthermore, for the properties portion of this section, Figure 4 in the appendix seems relevant to include in the main paper or at least mention it exists in the appendix. Figure 4 might benefit from some additional annotations. For example, what is the coordinate system in yellow exactly?
-In Definition 3.1, Is there something more going on in the integral that is missing? It seems to be an integral over a constant function of the variables.
-The authors should explain what a tree-like equivalence means because the paper mentions it several times.
-The related works section needs to be more cohesive in parts. Consider how each of the works cited ties to the author's work. For example, the first sentence in the paragraph on "Control and RL" seems entirely unrelated to the fact the authors consider value-based algorithms the most closely tied to their work.
-Equation 4.1 introduces expectations over \Omega but needs to explain what this space is. The closest we found was defining that "\omega" is a sample of the noise model. Is this the same noise as defined in section 3.2 for SDS systems? Figure 5 in the appendix suggests that \Omega is the distribution of trajectories. If so, this should be more obvious in the main paper.
-Several sections begin with a single sentence before jumping into the sub-sections. Either expand these sentences into meaningful paragraphs or delete them because they add limited additional information.
-Figure 2 Left is ambiguous as to what we should interpret from it. Neither the paper nor the caption provides helpful information for understanding the Figure.
These suggestions are non-exhaustive to the concerns we had with the writing. The reviewer's sense of the paper is that the authors want to focus on the generality of their framework, which hurts the paper as a whole. The authors writing will likely be more clear if they focus solely on:
1.) The path tracking problem and the challenges in solving this class of problems
how their new framework can absorb the value iteration framework and address its limitations when solving this problem
2.) Proposing their novel MPC algorithm based on the path Signature framework.
3.) Any more general discussion for future algorithms, such as novel reinforcement learning algorithms or justifying the generality of the framework, should then be put into future work.

The framework seems promising, but specifying the limitations of the Path Signatures approach would be good to discuss. One problem the reviewer imagines with this approach is using Kernels. Still, our experience with them is in the context of Gaussian Processes and their computation issues (e.g., inverting a matrix). Do kernels in path signatures suffer similarly? If more meaningful limitations exist, we suggest discussing them in the paper.

In terms of experiments, the reviewer would like clarification on the choice of baselines, which, at this time, we need more comparison to validate the improved performance of Signature MPC. From the author's citations, other researchers have studied the path-tracking problem previously. Why weren't any such algorithms included in the evaluation beyond MPC or model-free RL?
Particularly for MPC, although our personal experience is limited, isn't MPC a well-established algorithm in the literature? Why weren't other versions of MPC algorithms included in the experiments? We would like to have an explanation for this choice in the evaluation.
SAC as a baseline could have been clearer to the reviewer. From our understanding, MPC requires a dynamics model for a control problem. If that's the case, why even bother with model-free RL algorithms if there is a sufficiently accurate simulator? One could otherwise optimize SAC against this model and improve the robustness of SAC performance under disturbances done in your experiments. If there's evidence to the contrary that SAC can outperform MPC with a suitable simulator (which the results suggest isn't the case), that would be good to know.
Also, supposing that model-free RL algorithms are a good choice of baseline, what were the motivations to only compare against SAC? Publicly available repositories of Deep RL algorithms often make switching between algorithms when evaluating a task is easy. For the author's purposes, this would add further evidence of the benefits of their algorithm.

**Questions:**

1. Do path signatures need to be learned? It seems like only the policy is optimized and that the Signatures are known. If the signatures need to be known, this seems like a limitation or an assumption to state clearly.

2.What additional work would be needed to apply the signatures framework to problems different from path tracking? Is it plausible to assume one could derive Path Signature RL algorithms because of the connections with dynamic programming? What are the challenges of doing this?

3.Is the notion of way-points a means of partitioning a continuous trajectory into discrete steps to optimize over?

4.From our understanding, a challenge with path tracking is that, in practice, one needs a reference trajectory to optimize a policy to follow these paths. Is this correct, or is it a misunderstanding on the reviewer's part? The point of asking is it feels like a chicken and egg problem to expand the applications of your work to other problem settings (e.g., you would need the optimal value function to track to apply the Path signature framework).

---

### Official Review · Reviewer_AcwD · 2023-11-05

**Soundness:** 2 fair
**Presentation:** 1 poor
**Contribution:** 2 fair
**Rating:** 5
**Confidence:** 3

**Summary:**

In this paper, the authors combines path signature with MPC for dynamic control tasks. Unlike regular MPC where tracking control costs are defined usually in the L2 space and over some trivial properties of the trajectories, in this paper, the authors leverage the path signature, a more powerful series of scalar that can represent the trajectory properties. The authors show that, by defining the MPC cost over the signature space can improve the tracking accuracy in a simulated robot arm setup.

**Strengths:**

The strengths of the paper:

1) A novel formulation of MPC which optimizes the trajectory tracking problem over the signature space.
2) Better results compared with various baselines on robot arm tracking problems.

**Weaknesses:**

The weakness of the paper:

1) The paper is clearly written by authors from control/applied math background and also tailored for audiences of similar backgrounds. However, the readability of this paper is pretty low with the heavily-loaded math definitions for general researchers.

2) The signature MPC seems can only solve tracking problems: either following a path or stabilize the system.  Comparably, the baseline MPC, even with a lower tracking performance, can solve optimal control problems where there is no desired path to follow.

3) The signature MPC has additional computational costs to obtain the signatures. While the signature MPC may require less steps for converging, it is hard to tell if the wall clock time is actually less compared with a "slow" regular MPC.

4) If the claim is that signature MPC can be robust under system uncertainty, please include robust MPC such as tube MPC etc for a more comprehensive comparison.

**Questions:**

N/A